# MoReDrop: Dropout Without Dropping

## Abstract

Dropout has been instrumental in enhancing the generalization capabilities of deep neural networks across a myriad of domains. However, its deployment introduces a significant challenge: the model distributional shift between the training and evaluation phases. Previous approaches have primarily concentrated on regularization methods, invariably employing the sub-model loss as the primary loss function and updating the sub-model. Despite this, those methods continue to encounter a persistent distributional shift during evaluation, a consequence of the implicit expectation inherent to the evaluation process. In this study, we introduce an innovative approach, namely Model Regularization for Dropout (MoReDrop). MoReDrop effectively addresses distributional shift by prioritizing the loss function from the dense model, supplemented by a regularization term derived from the pair of dense-sub models. Importantly, We actively update only the dense model; the sub-model is passively updated due to shared attributes. This maintains the model consistency, i.e., the dense model, during training and evaluation, while the regularizer retains dropout benefits. To further mitigate the computational cost, we propose a lightweight version of MoReDrop, denoted as MoReDropL. This variant trades off a degree of generalization ability for reduced computational burden by employing dropout only at the last layer. Our experimental evaluations, conducted on several benchmarks across multiple domains, consistently demonstrate the scalability and efficiency of our proposed algorithms.

## 1 Introduction

In recent years, deep Neural Networks (DNNs) (Salakhutdinov, 2014; Schmidhuber, 2015) have made significant advancements across a wide range of areas such as computer vision, reinforcement learning, and natural language processing (Deng et al., 2009; Mnih et al., 2015; He et al., 2016; Vaswani et al., 2017; Ho et al., 2020; Jumper et al., 2021; Devlin et al., 2019). While DNNs hold great promise with deeper networks (He et al., 2016; Wang et al., 2022), the model complexity correspondingly escalates rapidly. This rapid escalation underscores the need for effective regularization techniques to mitigate overfitting and enhance the generalization capabilities of these deep models. Numerous strategies have been developed to tackle these challenges, with dropout (Hinton et al., 2012; Srivastava et al., 2014) gaining prominence due to its simplicity and efficacy extensively utilized in many recent AI breakthroughs (Dosovitskiy et al., 2021; Jumper et al., 2021; Ramesh et al., 2022). Dropout generally uses a Bernoulli-distributed mask applied to each layer prior to each training step, which also implies independently and randomly deactivating each neuron with probability $p$.

Dropout training mimics the ensemble model approach by implicitly integrating an exponential number of sub-models, characterized by randomly deactivated neurons, with shared parameters/weights; and performing model updates through the minimization of an approximate expected loss function (Hinton et al., 2012; Srivastava et al., 2014). However, it is non-trivial to explicitly assemble those sub-models, and a single model characterized by scaled parameters without dropout is employed for the practical evaluation period. It introduces a subtle but unignorable distributional shift between training and evaluation. A variety of regularizers have been proposed to mitigate this issue by implementing *sub-to-sub* regularization paradigm, which imposes constraints between pairs or multiple sub-models. The primary objective of these constraints is to ensure consistency across different sub-models, thereby maintaining the expectation of a unified, coherent model for evaluation. Examples of such regularizers include $L_2$ distance Zolna et al. (2018), Kullback-Leibler (KL) divergence for two random sub-models (Liang et al., 2021) or worse-case sub-models (Xia et al., 2023).

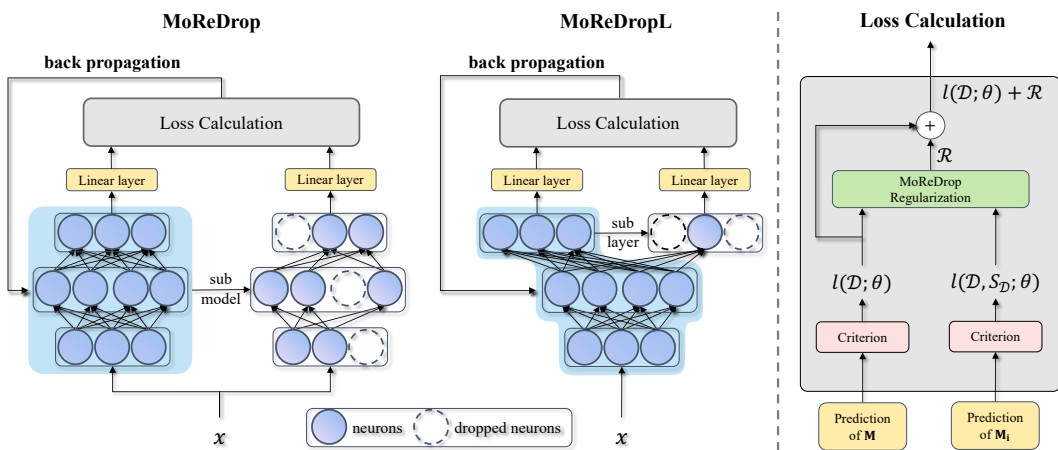

Figure 1: The overall framework and detailed loss computation of MoReDrop. **Left:** The input data $x$ is processed twice, once each through the dense model and its sampled sub-model, yielding two outputs. The loss function of the dense model is then regularized by minimizing the discrepancy between these two outputs. The gradient backpropagation is performed only on the dense model (blue-shaded area). **Right:** The details for calculating the regularized loss for the dense model.

Despite these efforts, sub-to-sub regularization approaches still present several limitations, such as introducing significant computational burdens and facing increasing challenges at high dropout rates due to the exponential proliferation of sub-models. Another line of research introduces the *dense-to-sub* regularization (Ma et al., 2017). This approach imposes constraints on the dense model and its corresponding sub-models throughout the training process, ensuring consistency across the pair of dense-sub models. However, all of the previous methods treat *the loss of sub-model* as the primacy loss function and a persistent distributional shift still arises from the expectation operation during evaluation.

In this study, we propose a novel dense-to-sub regularization approach, named Model Regularization for Dropout (MoReDrop), to mitigate the distributional shift by firstly employing *the loss of dense model* as the main loss function. Concretely, in each gradient step, every mini-batch dataset is passed through the network twice: for the dense model and its shared-parameter sub-models. MoReDrop then imposes a regularization term to the loss function from the output gap between the dense model and one of its sub-models. Significantly, we *actively* update only the dense model via gradient propagation, ensuring consistent model configuration, i.e., the dense model, for both training and inference. The regularizer preserves the generalization benefits of dropout.

Interestingly, the sub-models are *passively* updated through parameter sharing with the dense model in each iteration. However, in terms of computational load, MoReDrop necessitates an extra matrix multiplication for the whole sub-model. This is typically unfeasible for models with buried deep layers characterized by millions of parameters (Devlin et al., 2019; Dosovitskiy et al., 2021). To mitigate this issue, we introduce a lightweight version of MoReDrop, termed MoReDropL, which only applies dropout in the final layer with shared parameters, with the extra matrix multiplication arising solely from the *last-layer* of the sub-model. MoReDropL compromises a degree of generalization ability for lower computation burden.

We assess our proposed methods across various models and tasks, containing image classification and language understanding. Experiments show that our proposed methods allow for a higher dropout rate, which potentially further improves the performance but avoids the distributional shift. We also observe that MoReDrop consistently delivers superior performance compared to state-of-the-art baselines. Surprisingly, MoReDropL also surpasses previous methods in many tasks though it trades off the model generalization ability for computation efficiency.

## 2 PRELIMINARIES

**Notation** The training set, denoted as $\mathcal{D}$, consists of pairs $\{(x_1, y_1), \dots, (x_N, y_N)\}$, where $N$ signifies the total number of pairs in $\mathcal{D}$. In this context, each pair $(x_i, y_i)$ in $\mathcal{D}$ is typically considered

an independent and identically distributed (i.i.d.) sample drawn from the respective distributions of $X \in \mathcal{X}$ and $Y \in \mathcal{Y}$, respectively.

Consider a DNN, denoted by $\mathbf{M}$, consisting of $L$ hidden layers, with $X$ and $Y$ representing the input and output, respectively. Each layer in the network is indexed by $l$, which spans from 1 to $L$. The output vector from the $l^{th}$ layer is signified by $\mathbf{h}^{(l)}$. In this setup, the network's input is specified as $\mathbf{h}^{(0)} = x$, and the final network output is $\mathbf{h}^{(L)}$. The network $\mathbf{M}$ is characterized by a set of parameters collectively symbolized by $\theta = \{\theta_l : l = 1, \ldots, L\}$. Here, $\theta_l$ encapsulates the parameters associated with the $l^{th}$ layer. With slight abuse of notation, we indicate $l(\mathcal{D}; \theta)$ as the loss function.

**Dropout** In the naïve dropout formulation (Hinton et al., 2012; Srivastava et al., 2014), each layer is associated with $\Gamma^{(l)}$, a vector composed of independent Bernoulli random variables. Each of these variables has a probability $p_l$ of taking the value 0 and a probability $1 - p_l$ of assuming the value 1. This is analogous to independently deactivating the corresponding neuron (effectively setting each weight to zero) with a probability $p_l$. We introduce a set of dropout random variables, denoted by $S = \{\Gamma^{(l)} : l = 1, \ldots, L\}$, where $\Gamma^{(l)}$ corresponds to the dropout random variable for the $l^{th}$ layer. We can represent the deep neural network $\mathbf{M_i}$ as:

$$\mathbf{h}^{(l)} = f_l(\mathbf{h}^{(l-1)} \odot \gamma^{(l)}),$$

where $\odot$ denotes the element-wise product, and $f_l$ represents the transformation function for the $l^{th}$ layer. For instance, if the $l^{th}$ layer is a fully connected layer with a weight matrix $W$, a bias vector $b$, and a sigmoid activation function $\sigma(x) = 1/(1 + \exp(-x))$, then the transformation function is defined as $f_l(x) = \sigma(Wx+b)$. To justify the connection between the dense model and the sub-model, we represent $\mathbf{M_i}$ as the sub-model derived from $\mathbf{M}$ through the application of dropout, $i$ can be any number to represent different sub-models. We also use $\mathbf{h}^{(l)}(x, s; \theta)$ to denote the output of the $l^{th}$ layer given the input $x$ and the dropout value $s$, under the parameter set $\theta$.

Conceptually, dropout seeks to train an ensemble of exponentially many neural networks concurrently, with each network corresponding to a unique configuration of deactivated units, while sharing the same weights or parameters (Hinton et al., 2012; Hara et al., 2016). We denote the loss function with dropout as $l = (\mathcal{D}, S_{\mathcal{D}}; \theta)$, where $S_{\mathcal{D}} = \{S_1, \ldots, S_N\}$ represents the set of dropout variables.

**Model distributional shift.** As previously mentioned, dropout implicitly forms an ensemble of neural networks via weight sharing during the training phase. However, for evaluation, a deterministic dense model without dropped neurons is employed to approximate the ensemble operation. This approximation results in a model distributional shift, noted as $\mathcal{G}$, between training and evaluation when dropout is used:

$$\mathcal{G} = \mathbb{E}_S\left[\mathbf{H}^{(L)}(x, S; \theta)\right] - \mathbf{h}^{(L)}(x, \mathbb{E}[S]; \theta), \tag{1}$$

where the LHS of the minus signifies the ideal ensemble model with dropout for evaluation, which is represented by a dense model with expected activate units on the RHS of the minus.

## 3 Model Regularization for Dropout

In this section, we delve into the specifics of our proposed algorithm that operates the loss of dense function under dense-to-sub regularization, MoReDrop and MoReDropL, with the high-level structure depicted in Figure 1. We then proceed to discuss the chosen regularizer and the practical implementation. Finally, we draw a comparison between our algorithm and the previous methods to alleviate the model distributional shift, elucidating why our proposed algorithm exhibits superior performance, even under high dropout rates (as corroborated by experiments in Section 4).

### 3.1 Dense Loss with Dense-to-Sub Regularization

In supervised learning, for standard dense model training, the loss function is to minimize the following negative log-likelihood function:

$$l(\mathcal{D}; \theta) = -\sum_{i=1}^{N} \log p\left(y_i \mid x_i; \theta\right). \tag{2}$$

For dropout training, the loss function additionally incorporates the marginalization of the dropout variables (Wang & Manning, 2013; Srivastava et al., 2014):

$$\mathrm{E}_{S_\mathcal{D}}[l(\mathcal{D}, S_\mathcal{D}; \theta)] = \mathrm{E}_{S_\mathcal{D}}\Big[ -\sum_{i=1}^{N} \log p(y_i|x_i, S_i; \theta)\Big]. \tag{3}$$

The key contributor to the distributional shift is the *active* updating of dropout (sub-) models throughout the training phase. This procedure implicitly establishes a parameter-sharing ensemble model Ma et al. (2017); Liang et al. (2021). However, during the inference stage, a deterministic model is employed. To address this discrepancy, we propose a novel regularization method that transitions from dense to sparse models, termed Model Regularization for Dropout (MoReDrop). Specifically, we prioritize the loss derived from the dense model without dropout (Equation (2)) and perform gradient backpropagation only on the dense model, which ensures the model configuration consistency, i.e., the dense model, is applied throughout both the training and inference stages. Furthermore, we harness the advantages of the dropout models by imposing constraints on the pair of the dense model $\mathbf{M}$ and a specific shared-parameter sub-model $\mathbf{M_i}$ with *passive* updating without gradient backpropagation, courtesy of their shared-parameter nature with the dense model. To further elaborate on how MoReDrop addresses the distributional shift, we make a comprehensive comparison with previous work, shown in Appendix Table 4.

In Theorem 3.1, we demonstrate that the loss function from $\mathbf{M_i}$ provides a point-wise upper bound for the loss function from $\mathbf{M}$ (see details in Appendix B). Theorem 3.1 allows for representing the regularization with the first-order moment:

$$\mathcal{R} = g(\mathbb{E}_{S_\mathcal{D}}[l(\mathcal{D}, S_\mathcal{D}; \theta)] - l(\mathcal{D}; \theta)]),$$

where $g(\cdot)$ is a monotonically increasing function.

**Theorem 3.1.** *The expected loss function of standard dropout Equation* (3) *is an upper bound for the standard loss function without dropout Equation* (2):

$$l(D; \theta) \leq \mathbb{E}_{S_\mathcal{D}}[l(\mathcal{D}, S_\mathcal{D}; \theta)].$$

In the present study, we adopt the function $g = (\exp(\alpha \cdot x) - 1)/(\exp(\alpha \cdot x) + 1)$, a variant of the Logistic Sigmoid function to confine the output within the range $[-1, 1]$, where $\alpha$ serves as a weight that scales the first-order moment function. This particular formulation of the regularization term exhibits two primary characteristics. First, it facilitates a unified hyperparameter search space across varied tasks, thereby substantially diminishing the search space. Second, its bounded nature imparts robustness to the loss function against varying dropout rates $p$, rendering it possible to train a near-optimal model even at high values of $p$. We perform an ablation study on various loss functions that might not align with the aforementioned characteristics, as detailed in Section 4, which demonstrates the superiority of the loss function $g$. However, it is important to acknowledge the possibility of other, more effective loss functions, which we aim to explore in future research. Specifically, the regularization term we employed is as follows:

$$\mathcal{R} = \frac{\exp(\alpha \cdot (\mathbb{E}_{S_\mathcal{D}}[l(\mathcal{D}, S_\mathcal{D}; \theta)] - l(\mathcal{D}; \theta))) - 1}{\exp(\alpha \cdot (\mathbb{E}_{S_\mathcal{D}}[l(\mathcal{D}, S_\mathcal{D}; \theta)] - l(\mathcal{D}; \theta))) + 1}, \tag{4}$$

and the final optimization objective in our algorithm is:

$$\operatorname*{argmin}_{\theta} -\sum_{i=1}^{N} \log p(y_i \mid x_i; \theta) + \mathcal{R}, \tag{5}$$

where it is bifurcated into two principal elements: the primary one being the loss of the dense model, which guarantees consistency; and a supplementary regularization term $\mathcal{R}$, incorporated to draw upon the advantages offered by dropout, particularly generalization abilities. Importantly, MoReDrop conducts gradient backpropagation exclusively on the dense model, yet it still capitalizes on the advantages of dropout models.

To further mitigate computational costs, we introduce a light variant of MoReDrop, termed as MoReDropL. This version retains the guiding principle of MoReDrop, where the main loss comes

from the dense model regularized by the interplay between the dense model and its sub-models. The key distinction between MoReDropL and MoReDrop lies in their network structure for utilizing dropout: MoReDropL employs dropout solely in the *final* layer, thereby circumventing additional matrix computations, while MoReDrop applies dropout across *all* layers, necessitating extra matrix computations for all networks. Although MoReDropL sacrifices a degree of generalization capability, this compromise enables a significant reduction in computational burden.

## 3.2 ALGORITHM SUMMARY

During every gradient update, we execute the forward operation twice on the model using the same randomly sampled mini-batch dataset: once for the dense model and once for its shared-parameter sub-model employing dropout. Subsequently, we calculate the loss function based on these two forward operations. We summarize our final algorithm in Algorithm 1. Note that we only apply gradient update to the dense model $\mathbf{M}$; the shared-parameter sub-model $\mathbf{M_i}$ does not undergo updates through gradient backpropagation and we note the stop gradient operator as $\square[\cdot]$. Further, we note that only one explicit sub-model is sampled, which represents an implicit ensemble of exponentially shared parameter networks (Ma et al., 2017; Liang et al., 2021). We also provide a Pseudocode with a PyTorch-like style in Appendix A.

---

**Algorithm 1** Model Regularization for Dropout

---

**Require:** Training dataset $\mathcal{D} = \{(x_i, y_i)\}_{i=1}^N$, weight $\alpha$.
 1: Initialize $\mathbf{M}_\theta$
 2: **for** $t = 1, 2, \cdots, N$ **do**
 3:     Randomly sample mini-batch $B_i \sim \mathcal{D}$
 4:     Forward the $B_i$ to the dense model $\mathbf{M}_\theta$ and obtain $l(\mathcal{D}; \theta)$ via Equation (2)
 5:     Forward the $B_i$ to the sub-model $\mathbf{M_{i\theta}}$ and obtain $\square\mathbb{E}_{S_\mathcal{D}}[l(\mathcal{D}, S_\mathcal{D}; \theta)]$ via Equation (3)
 6:     Update $\mathbf{M}_\theta$ by Equation (5)
 7: **end for**

---

## 3.3 DISCUSSION

Our proposed method shares commonalities with several existing works, especially those focusing on sub-to-sub regularization (Zolna et al., 2018; Liang et al., 2021; Xia et al., 2023), as well as dense-to-sub regularization (Ma et al., 2017). Both MoReDrop and MoReDropL are categorized within the dense-to-sub regularization framework. However, in contrast to prior methods where the primary loss function originates from the sub-model, our approach sets the primary loss function based on the dense model. This key differentiation offers our method several significant advantages: (1). *Robustness to High Dropout Rates*: Our methods demonstrate resilience to high dropout ratios. This is attributable to the fact that the priority optimization target in our method, i.e., the dense model loss function, maintains no dependency on dropout ratios. (2). *Mitigation of Model Distributional Shift*: The primary loss function in our method explicitly ensures consistency during both model training and evaluation phases. The regularization term leverages the benefits of dropout from its sub-model, without the need for explicitly dropping neurons during the gradient update process in neural networks. (3). *Computational efficiency*: MoReDropL restricts additional matrix manipulation to the final layer alone. In contrast, other methods, including MoReDrop, apply this additional matrix manipulation across the entire model.

We explore these benefits through our experiments, as detailed in Section 4. Moreover, the dense-to-sub regularization approach in our methods facilitates stable training and rapid convergence speed due to the significantly reduced uncertainty associated with the dense model (without dropout), particularly in the context of high dropout ratios. Specifically, for high dropout ratios, sub-to-sub regularization methods face challenges due to the exponential growth and combination of sub-models. This escalating difficulty for sub-to-sub regularization may result in misguided optimization priorities, managing to maintain consistency among sub-models but failing to enhance the final performance. We provide a systematic analysis of this in Section 4.3.

## 4 EXPERIMENTS

To underscore the wide-ranging applicability of our proposed method, we conducted a thorough evaluation spanning distinct machine learning domains, i.e., image classification, and general language understanding, with different backbones algorithms (shown in Section 4.1 and Section 4.2). Next, we conduct a detailed loss analysis and compare MoReDrop with R-Drop, which provides a crucial explanation for the superior performance of MoReDrop, as discussed in Section 4.3. In Appendix C.4.1, we showcase the robustness of MoReDrop under different hyperparameter combinations, even in scenarios with extremely high dropout rates. More details of experimental settings for each dataset and backbone algorithm can be found in Appendix C. Lastly, in Section 4.4, we present the training time of our proposed algorithms in comparison with various baselines in Appendix Table 3, demonstrating the training efficiency of MoReDrop.

### 4.1 IMAGE CLASSIFICATION DOMAINS

**Benchmark Datasets.** Our image classification experiments were conducted on three well-recognized benchmark datasets: `CIFAR-10`, `CIFAR-100` (Krizhevsky et al., 2009) and `ImageNet` (Deng et al., 2009). The `CIFAR-10` and `CIFAR-100` datasets both consist of low-dimensional pixel images, with the primary distinction between them being the number of categories they feature, as indicated by their respective names. Conversely, the `ImageNet` dataset presents a significantly greater challenge, encompassing more than $1,000$ categories.

**Model & Training.** To offer a scalable comparison of MoReDrop in the domain of image classification, we utilize two distinct models: a smaller model with $1.2$ million parameters (ResNet-18) (He et al., 2016) and a larger model with 86 million parameters (ViT-B/16) (Dosovitskiy et al., 2021). Note that the standard ResNet-18 does not incorporate dropout, while the default configuration of the vanilla ViT-B/16 utilizes a dropout rate of $p = 0.1$. For the ResNet-18, our baselines comprise: (1) DropBlock (Ghiasi et al., 2018), which mitigates overfitting by dropping continuous regions of neurons, and (2) DropPath (Larsson et al., 2017),

Table 1: Accuracy on CIFAR-10, CIFAR-100 and ImageNet. Both MoReDrop and MoReDropL consistently outperform the baseline across all tasks.

| Methods | CIFAR-10 | CIFAR-100 | ImageNet |
|---|---|---|---|
| ResNet-18 | $95.44_{\pm0.07}$ | $77.78_{\pm0.07}$ | - |
| + DropPath | $95.35_{\pm0.05}$ | $78.12_{\pm0.11}$ | - |
| + DropBlock | $95.53_{\pm0.12}$ | $78.72_{\pm0.06}$ | - |
| + MoReDropL | $\mathbf{95.79}_{\pm0.21}$ | $79.11_{\pm0.05}$ | - |
| + DropPath + MoReDrop | $\mathbf{95.60}_{\pm0.14}$ | $79.25_{\pm0.19}$ | - |
| + DropBlock + MoReDrop | $\mathbf{96.41}_{\pm0.11}$ | $79.53_{\pm0.32}$ | - |
| ViT-B/16 | $98.68_{\pm0.24}$ | $92.78_{\pm0.10}$ | $84.05_{\pm0.15}$ |
| + R-Drop | $98.97_{\pm0.01}$ | $92.90_{\pm0.02}$ | $84.16_{\pm0.04}$ |
| + MoReDropL | $\mathbf{99.14}_{\pm0.03}$ | $\mathbf{93.25}_{\pm0.03}$ | $\mathbf{84.62}_{\pm0.12}$ |
| + MoReDrop | $\mathbf{99.10}_{\pm0.06}$ | $\mathbf{93.38}_{\pm0.04}$ | $\mathbf{84.43}_{\pm0.06}$ |

which zeroes out an entire branch in the neural network during training, aiming to achieve the same goal as DropBlock. Note that ResNet incorporates batch normalization as a technique to combat overfitting, disrupting the training process (Ioffe & Szegedy, 2015). We restrict our comparison to using MoReDropL within the context of last-layer dropout only. As for the ViT-B/16 algorithm, we incorporate R-Drop (Liang et al., 2021) with the aim of alleviating model distributional shift, a goal akin to that of MoReDrop. For both ResNet-18 and ViT-B/16, MoReDropL only utilizes standard dropout before the last linear layer as proposed in Figure 1. In all baselines, we set the dropout rate to $0.1$, as recommended by the original papers, and this rate has been found to yield the best performance compared to other dropout rates in our evaluation, as shown in Appendix Table 6.

**Results.** The results displayed in Table 1 represent averages obtained from 5 independent seeds. For MoReDrop, we sweep the parameters $p$ and $\alpha$ across the sets $\{0.1, 0.2, 0.3, 0.4, 0.5, 0.7, 0.9\}$ and $\{0.1, 0.5, 1, 2\}$, respectively. The best performance of MoReDrop is then presented in Table 1. Experimental details are shown in Appendix C.2. When integrated with DropPath and DropBlock for the ResNet-18 model, MoReDrop consistently delivers superior performance on both the `CIFAR-10` and the more challenging `CIFAR-100` datasets. Notably, when paired with DropBlock, MoReDrop realizes a significant increase in accuracy compared to both the original ResNet-18 and DropBlock, with improvements of approximately $1\%$ in the `CIFAR-10` dataset and $1.8\%$ in the more challenging

Table 2: The results of NLU tasks on the GLUE benchmark. MoReDrop outperforms the backbone model for all tasks and MoReDropL outperforms the backbone in 10 out of 16 tasks.

| Methods | CoLA Matt. | SST-2 Acc. | MRPC Acc./F1 | STS-B P. Corr. | QQP Acc./F1 | MNLI m./mm. | QNLI Acc. | RTE Acc. | Average |
|---|---|---|---|---|---|---|---|---|---|
| BERT-base | $56.49_{\pm0.24}$ | $93.31_{\pm0.12}$ | $85.10_{\pm0.31}$ / $89.41_{\pm0.18}$ | $87.92_{\pm0.22}$ | $91.38_{\pm0.02}$ / $87.55_{\pm0.02}$ | $83.49_{\pm0.16}$ / $84.84_{\pm0.18}$ | $91.46_{\pm0.12}$ | $67.99_{\pm0.21}$ | 83.54 |
| + MoReDropL | $58.23_{\pm0.39}$ | $92.52_{\pm0.07}$ | $87.12_{\pm0.22}$ / $90.82_{\pm0.29}$ | $88.24_{\pm0.18}$ | $91.21_{\pm0.04}$ / $87.77_{\pm0.09}$ | $83.97_{\pm0.11}$ / $84.46_{\pm0.11}$ | $91.14_{\pm0.12}$ | $69.05_{\pm0.13}$ | 84.05 |
| + MoReDrop | $\mathbf{58.99}_{\pm0.26}$ | $\mathbf{93.53}_{\pm0.10}$ | $\mathbf{87.18}_{\pm0.31}$ / $\mathbf{90.86}_{\pm0.22}$ | $\mathbf{88.31}_{\pm0.09}$ | $\mathbf{91.41}_{\pm0.04}$ / $\mathbf{87.97}_{\pm0.04}$ | $\mathbf{84.98}_{\pm0.21}$ / $\mathbf{85.27}_{\pm0.22}$ | $\mathbf{91.59}_{\pm0.14}$ | $\mathbf{69.98}_{\pm0.27}$ | **84.55** |
| RoBERTa-base | $60.07_{\pm0.22}$ | $93.86_{\pm0.28}$ | $87.50_{\pm0.33}$ / $90.84_{\pm0.21}$ | $89.68_{\pm0.22}$ | $91.02_{\pm0.07}$ / $87.40_{\pm0.11}$ | $87.77_{\pm0.11}$ / $87.49_{\pm0.24}$ | $92.62_{\pm0.07}$ | $72.77_{\pm0.33}$ | 85.55 |
| + MoReDropL | $\mathbf{62.39}_{\pm0.31}$ | $94.09_{\pm0.41}$ | $88.19_{\pm0.54}$ / $91.52_{\pm0.33}$ | $90.46_{\pm0.27}$ | $91.38_{\pm0.05}$ / $87.94_{\pm0.18}$ | $87.20_{\pm0.13}$ / $86.71_{\pm0.27}$ | $92.27_{\pm0.02}$ | $\mathbf{79.48}_{\pm1.33}$ | 86.51 |
| + MoReDrop | $62.37_{\pm0.33}$ | $\mathbf{94.79}_{\pm0.37}$ | $\mathbf{89.80}_{\pm0.22}$ / $\mathbf{92.44}_{\pm0.16}$ | $\mathbf{90.55}_{\pm0.16}$ | $\mathbf{91.55}_{\pm0.09}$ / $\mathbf{88.17}_{\pm0.09}$ | $\mathbf{87.90}_{\pm0.10}$ / $\mathbf{87.60}_{\pm0.17}$ | $\mathbf{92.73}_{\pm0.11}$ | $77.45_{\pm0.41}$ | **86.85** |

`CIFAR-100` dataset. The consistently improved performance across different dropout methods, i.e., DropPath and DropBlock, attests to the general applicability of MoReDrop.

Besides, with the backbone of ViT-B/16, we find that MoReDrop outperforms the vanilla model and its variant with R-Drop over three degrees of challenges tasks. The small margin gained ($< 1\%$) from MoReDrop, compared with the backbone of ResNet-18, is attributed to the saturated performance by its ViT-B/16 backbone. Surprisingly, MoReDropL even outperforms MoReDrop in the `CIFAR-10` dataset by $0.04\%$ points. This might be due to the loss of plasticity in pre-trained models, where further training has a limited impact on most neurons (Achille et al., 2017; Zilly, 2022).

Comparable advancements are also evident in the context of the more challenging large-scale dataset, `ImageNet`. We observe that R-Drop necessitates approximately 3x the number of training epochs to converge, yet its final performance is not on par with our methods (both MoReDrop and MoReDropL). This observation underscores the superiority of our proposed algorithm on challenging tasks.

Our proposed algorithm, MoReDrop, consistently exhibits robustness to hyperparameters across various tasks, as shown in Appendix C.4.1. Notably, it achieves optimal performance even at dropout rates exceeding $0.1$, a threshold that often hampers the performance of other methods. This observation underscores the potential for high generalization ability in models with elevated dropout rates, and emphasizes the resilience of our algorithm under such conditions.

## 4.2 NATURAL LANGUAGE UNDERSTANDING

**Benchmark Datasets.** To broaden our evaluation to encompass natural language understanding, we assess our proposed methods on the standard development sets of the General Language Understanding Evaluation (`GLUE`) benchmark (Wang et al., 2019). The GLUE benchmark includes eight unique tasks, all of which involve text classification or regression. The distinct characteristics of each task, such as the nature and complexity of the language understanding challenge, as well as the volume and diversity of the training data, provide a comprehensive and robust testing ground for our proposed methodology. The evaluation metrics for the eight tasks and experiment details are shown in Appendix C.3.

**Model & Training.** We utilize two publicly available pre-trained models: BERT-base (Devlin et al., 2019) and RoBERTa-base (Liu et al., 2019) as our foundation for fine-tuning. As both BERT-base and RoBERTa-base employ standard dropout, we could directly apply our method and use the original models for comparison. To ensure a fair comparison, we retained the same training hyperparameters setting as in the original models.

**Results.** We present the final performance in Table 2 averaged by 5 independent seeds. The sweeping sets of hyperparameters are the same as the image classification domains. MoReDrop consistently outperforms the baselines across all tasks, showing improvements of approximately $1\%$ and $1.3\%$ on BERT-base and RoBERTa-base models, respectively. We also find that the average performance of MoReDropL exhibits improvements over the baseline by approximately $0.5\%$ for the BERT-base and $1\%$ for RoBERTa-base models, even with minimal modifications to the last layer using dropout. Intriguingly, MoReDropL surpasses the performance of MoReDrop on certain tasks, most notably achieving a performance margin of $2\%$ on the `RTE` task compared to MoReDrop. We attribute this to the phenomenon of the loss of plasticity, as we discussed in Section 4.1.

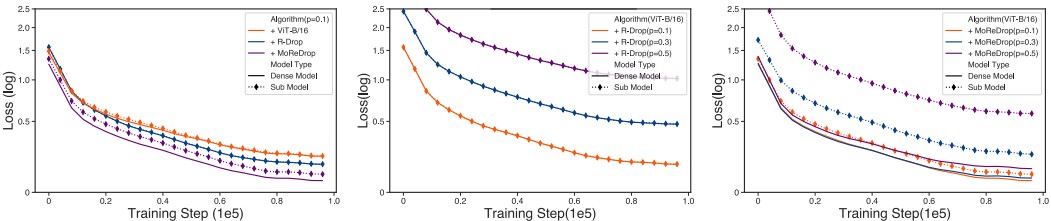

Figure 2: The training loss curves over ViT/16, comparing different dropout methods and rates on the CIFAR-10 dataset. **Left:** Training loss curves of various methods with a consistent dropout rate ($p = 0.1$). **Middle:** Training loss curves of R-Drop under varying dropout rates. **Right:** Training loss curves of MoReDrop under different dropout rates.

### 4.3 LOSS ANALYSIS

In this section, we empirically validate the limitations mentioned of prior approaches in Section 3.3 through a comprehensive comparative analysis between MoReDrop and the state-of-the-art model regularization approach, R-Drop. In Figure 2, we present the training loss curves over ViT-B/16, comparing different dropout methods and rates on the `CIFAR-10` dataset. Note that for the sub-model loss from MoReDrop and the dense-model loss from R-Drop, we merely compute these losses, abstaining from conducting gradient updates.

As demonstrated in Figure 2 (Left), with an identical low dropout rate (as recommended by ViT-B/16), R-Drop attains a lower training loss and superior performance (as shown in Table 1). This holds true for both the dense model (blue solid line) and one of its sampled sub-models (blue dotted line), when compared to the vanilla algorithm (orange solid line). Notably, the training loss for both the dense (purple solid line) and the sub-model (purple dotted line) from MoReDrop are lower than those of R-Drop, aligning with higher performance as shown in Table 1.

Interestingly, we observe a significantly larger loss gap[1] between the dense model and the sub-model losses for MoReDrop compared to R-Drop in the training phase. The loss gap signifies the distance in the function space, serving as an approximation of the model discrepancy. This finding implies that even under a small dropout ratio, the purpose of maintaining high consistency among sub-models (approximating model distributional shift) may compromise the model expressivity, yielding sub-optimal performance relative to MoReDrop. Furthermore, despite these efforts, a trivial model distributional shift persists in R-Drop due to the inherent expectation operator.

To delve deeper into this, we present the loss curves from both dense and sub-models across different dropout rates of R-Drop in Figure 2 (Middle). We observe that the loss gap approaches zero irrespective of the dropout rate. However, this is concurrent with decreased performance, shown in Appendix Table 6, suggesting a strong performance compromise while constraining sub-model pairs.

In contrast, MoReDrop shows a larger loss gap as the dropout probability $p$ increases during training, while maintaining near-optimal final performance despite variations in $p$, as illustrated in Figure 2 (Right). This implies that MoReDrop does not compromise model generalization to accommodate constraints, leading to improved performance and robustness, even under high dropout rates, as illustrated in Figure 4. The improved performance of MoReDrop can be attributed to the primary dense model loss function, which effectively mitigates the model distributional shift, as well as the dense-to-sub regularization. This approach not only preserves the full potential of model expressivity but also capitalizes on the benefits provided by the dropout mechanism.

### 4.4 TRAINING TIME

In Table 3, we present the per epoch training time for MoReDrop, MoReDropL, and baselines on the `CIFAR-10` task, executed on a GPU (NVIDIA A800 80GB PCIe) using a batch size of 32. MoReDropL exhibits a significant efficiency advantage with nearly the same training time as the

---

[1]The loss gap in R-Drop approximates the model distributional shift in Equation (1). Conversely, in MoReDrop, the loss gap does not mirror this distributional shift due to its primary loss function is rooted in the dense model, consistently applied during training and evaluation.

backbone, and approximately 50% faster than both MoReDrop and R-Drop. For example, in the case of ViT-B/16, R-Drop and MoReDrop require 176s and 172s respectively to execute a training epoch, whereas MoReDropL requires only 92s, almost on par with the $90s$ needed for ViT-B/16.

## 5 RELATED WORK

**Dropout and its Variants.** Regularization plays a pivotal role in preventing overfitting in deep learning and large-scale models. A multitude of regularization techniques has been proposed to address this issue, including but not limited to weight decay, dropout, batch normalization, noise addition, early stopping, and label smoothing (Simonyan & Zisserman, 2015; Ioffe & Szegedy, 2015; Poole et al., 2014; Yao et al., 2007; Szegedy et al., 2016). Among these, dropout (Hinton et al., 2006) stands out as a particularly effective method due to its simplicity and broad applicability in different domains. For different model architectures, different dropout methods a variety of dropout methods have been proposed. These consist of DropConnect (Wan et al., 2013) for fully connected layers, SpatialDropout (Tompson et al., 2015) and DropBlock (Ghiasi

Table 3: Training time per epoch.

| Model | Training time (seconds) |
|---|---|
| ResNet-18 | 14 |
| + MoReDropL | 15 |
| + MoReDrop | 29 |
| ViT-B/16 | 90 |
| + R-Drop | 176 |
| + MoReDropL | 92 |
| + MoReDrop | 172 |

et al., 2018) for convolutional neural Networks, DropPath (Larsson et al., 2017) for ResNet, and DropHead (Zhou et al., 2020) for Transformer models. In addition to its role as a regularization method to prevent overfitting, dropout has also been utilized as a data augmentation technique (DeVries & Taylor, 2017; Zhong et al., 2020), further contributing to its effectiveness and versatility.

**Model Distributional Shift.** Prior work has revealed that dropout brings the inconsistency between training and inference stages, specifically, the model distributional shift. There are primary two categories to address this issue: (1) Sub-to-sub regularization paradigm aims to maintain the consistency between a pair of sub-models in the training process. In this paradigm, Fraternal Dropout (FD) (Zolna et al., 2018) employs $L_2$ distance on hidden states, R-Drop (Liang et al., 2021) utilizes the on two sampled sub-models with dropout, and Worst-Case Drop Regularization (WordReg) (Xia et al., 2023) holds the same inspiration with R-Drop but firstly find the two worse-case sub-models. (2) dense-to-sub regularization paradigm, i.e., Expectation Linear Dropout (ELD) (Ma et al., 2017), which maintains the consistency between a pair of dense-sub models as training progresses. MoReDrop belongs to the dense-to-sub regularization paradigm while still holding a significant difference.

## 6 CONCLUSION AND LIMITATIONS

In this study, we propose a simple yet effective approach, MoReDrop, to mitigate model distributional shift in dropout models while leveraging the benefits of dropout, without explicitly dropping neurons during gradient backpropagation. Specifically, the primary loss function in MoReDrop originates from the dense model and is regularized by the model gap approximation between the dense model and one of its sampled sub-models. To further reduce computational load, we introduce a lightweight version of MoReDrop, termed MoReDropL, which only performs matrix multiplication in the final layer, unlike MoReDrop which applies it across all layers, albeit at the cost of some generalization ability. Our experimental results across a variety of tasks and domains consistently demonstrate that both MoReDrop and MoReDropL achieve state-of-the-art performance in the majority of tasks. Interestingly, MoReDropL even outperforms MoReDrop in performance on the RTE task from the GLUE benchmark with a great margin. Our aspiration is that these insights will catalyze further exploration into creating neural network regularizers to manage model distributional shifts within dropout models. One potential limitation of this study pertains to the scalability of MoReDrop in more challenging domains, such as self-supervised learning and reinforcement learning. These areas have not been exhaustively investigated and often necessitate extensive domain-specific designs for neural networks.

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

Table 4: Comparison of different regularization approaches to mitigate the model distributional shift between the training and inference stages. It is important to highlight that the distributional gap between the models in the inference and training stages is 0 because to the same model configuration, i.e., the dense model, is used for both training and inference.

| Algorithms | Gradient back-propagation | Dense model update | Sub-model update | Regularizer | Distribution shift |
|---|---|---|---|---|---|
| FD (Zolna et al., 2018) | Sub-model | Passive | Active | Sub-to-sub | $\mathcal{G}$ |
| R-Drop (Liang et al., 2021) | Sub-model | Passive | Active | Sub-to-sub | $\mathcal{G}$ |
| WordReg Xia et al. (2023) | Sub-model | Passive | Active | Sub-to-sub | $\mathcal{G}$ |
| ELD (Ma et al., 2017) | Sub-model | Passive | Active | Dense-to-sub | $\mathcal{G}$ |
| MoReDrop (ours) | Dense model | Active | Passive | Dense-to-sub | 0 |

## A    PSEUDOCODE OF MOREDROP IN A PYTORCH-LIKE STYLE.

---

**Algorithm 2** Pseudocode of MoReDrop in a PyTorch-like style.

---

```
1  # M: dense model
2  # p: dropout rate
3  # alpha: regularization parameter
4  # g: regularization loss fuction
5  for (x,y) in loader: # load a minibatch
6
7      # activate dropout layer with dropout rate p
8      set_rate(p)
9      logits_sub = M(x) # logits of M_i
10
11     # shutdown dropout layer
12     set_rate(0)
13     logits_dense = M(x)  # logits of M
14
15     # calculate the loss gap
16     loss_dense = CrossEntropyLoss(logits_dense, y)
17     loss_sub = CrossEntropyLoss(logits_sub, y)
18     loss_gap = loss_sub - loss_dense
19     loss = loss_dense + g(alpha * loss_gap)
20
21     # dense model update
22     loss.backward()
23     update(M)
```

---

## B    PROOF OF THEOREM 3.1

*Proof.* For a tractable approximation for dropout variable $p$, we use Bayes' rule to express the parameterized conditional probability of the output $y$ given the input $x$ and $p$ under parameter $\theta$:

$$p(y \mid x; \theta) = \int_{\mathcal{S}} p(y \mid x, s; \theta) p(s) d\mu(s).$$

We then can rewrite the loss function for sub-models with dropout (Eqn. 3) as:

$$
\begin{aligned}
\mathrm{E}_{S_{\mathcal{D}}}[l(\mathcal{D}, S_{\mathcal{D}}; \theta)] &= -\int_{\mathcal{S}} \prod_{i=1}^{N} p(s_i) \Big( \sum_{i=1}^{N} \log p(y_i | x_i, s_i; \theta) \Big) d\mu(s_1) \dots d\mu(s_N) \\
&= -\sum_{i=1}^{N} \int_{\mathcal{S}} p(s_i) \log p(y_i | x_i, s_i; \theta) d\mu(s_i).
\end{aligned}
$$

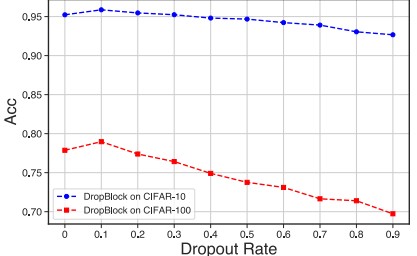 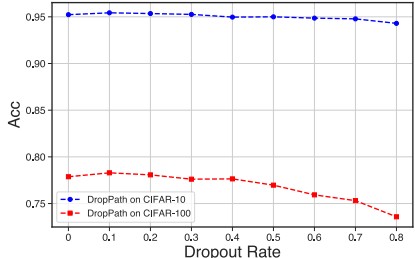

Figure 3: The accuracy of ResNet-18 employing DropBlock and DropPath under varying dropout rates on CIFAR-10 and CIFAR-100. **Left:** Accuracy curves of ResNet-18 utilizing DropBlock. **Right:** Accuracy curves of ResNet-18 utilizing DropPath. Note that DropPath failed to train the model under the dropout rate $p = 0.9$, hence this data point is not represented in the figure.

Given that $\log(\cdot)$ is a concave function, according to Jensen's Inequality:

$$\int_{\mathcal{S}} p(s) \log p(y|x, s; \theta) d\mu(s) \leq \log \int_{\mathcal{S}} p(s) p(y|x, s; \theta) d\mu(s).$$

Thus

$$\mathrm{E}_{S_{\mathcal{D}}}[l(\mathcal{D}, S_{\mathcal{D}}; \theta)] \geq \sum_{i=1}^{N} \log \int_{\mathcal{S}} p(s_i) p(y_i|x_i, s_i; \theta) d\mu(s_i) = l(\mathcal{D}; \theta).$$

$\square$

## C    EXPERIMENTS DETAILS

### C.1    HARDWARE SETUP

Our experiments were performed using PyTorch and run on NVIDIA GeForce RTX 3090 and NVIDIA A800 80GB PCIe graphics cards. For the `CIFAR-10` and `CIFAR-10` tasks, we utilized single-card training on the NVIDIA GeForce RTX 3090. For the `ImageNet` task, we employed distributed training across $4\times$ NVIDIA A800 80GB PCIe cards.

### C.2    IMAGE CLASSIFICATION

**Model Details.**    Our image classification experiments employed two distinct models: ResNet-18 and ViT-B/16. The former, ResNet-18, is a smaller model boasting 1.2 million parameters. As a member of the ResNet family, it utilizes a residual learning framework to streamline the training of networks, encompassing 18 layers that include convolutional, identity, and fully connected layers. Conversely, ViT-B/16 is a larger model, with 86 million parameters. This model is a variant of the Vision Transformer (ViT) model, which repurposes transformers, initially designed for natural language processing tasks, for computer vision tasks.

Table 5: Hyperparameters for Image Classification tasks.

| Model | $p$ | | | $\alpha$ | | | Batch Size | | | Epochs | | |
|---|---|---|---|---|---|---|---|---|---|---|---|---|
| | CIFAR-10 | CIFAR-100 | ImageNet | CIFAR-10 | CIFAR-100 | ImageNet | CIFAR-10 | CIFAR-100 | ImageNet | CIFAR-10 | CIFAR-100 | ImageNet |
| ResNet-18 | - | - | - | - | - | - | 128 | 128 | - | 200 | 200 | - |
| + DropPath | 0.1 | 0.1 | - | - | - | - | 128 | 128 | - | 200 | 200 | - |
| + DropBlock | 0.1 | 0.1 | - | - | - | - | 128 | 128 | - | 200 | 200 | - |
| + MoReDropL | 0.1 | 0.3 | - | 0.1 | 0.5 | - | 128 | 128 | - | 200 | 200 | - |
| + DropPath + MoReDrop | 0.5 | 0.1 | - | 0.1 | 0.1 | - | 128 | 128 | - | 200 | 200 | - |
| + DropBlock + MoReDrop | 0.3 | 0.2 | - | 0.5 | 0.5 | - | 128 | 128 | - | 200 | 200 | - |
| ViT-B/16 | 0.1 | 0.1 | 0.1 | - | - | - | 32 | 256 | 64 | 50 | 50 | 10 |
| + R-Dorp | 0.1 | 0.1 | 0.1 | 0.3 | 0.3 | 0.3 | 32 | 256 | 64 | 50 | 50 | 30 |
| + MoReDropL | 0.1 | 0.4 | 0.1 | 1 | 1 | 1 | 32 | 256 | 64 | 50 | 50 | 10 |
| + MoReDrop | 0.1 | 0.1 | 0.1 | 1 | 0.5 | 1 | 32 | 256 | 64 | 50 | 50 | 10 |

**Baseline Setting & Experiments Design.** While the vanilla ResNet-18 does involve the dropout technique, we take DropBlock (Ghiasi et al., 2018) and DropPath (Larsson et al., 2017) as our baselines. DropBlock mitigates overfitting by dropping continuous regions of neurons, while Drop-Path zeroes out an entire branch in the neural network during training, as opposed to just a single unit, making it a perfect match for ResNet. We set the dropout rate to $0.1$, as recommended by the original papers for both DropBlock and DropPath, as well as the best dropout rate referring in our experiments(as shown in Figure 3). This rate has been determined to yield the best performance in comparison to other dropout rates. Further details about these experiments can be found in Table 5.

In contrast, we adopt R-Drop as comparison in ViT-B/16, which minimizes the bidirectional KL-divergence of the output distributions of any pair of sub models sampled from dropout in model training. Given that ViT-B/16 incorporates standard dropout, MoReDrop can be directly applied to this model.

Meanwhile, MoReDropL was applied before the last linear layer, as well as the classifier with standard dropout for both ResNet-18 and ViT-B/16.

Table 6: Comparison of various algorithms on the CIFAR-10 task under different dropout rates.

| Dropout Rate | ViT-B/16 | R-Drop | MoReDrop |
|---|---|---|---|
| $p = 0.1$ | 98.72 | 98.98 | 99.11 |
| $p = 0.3$ | 98.66 | 98.68 | 99.04 |
| $p = 0.5$ | 97.80 | 97.52 | 98.85 |
| $p = 0.7$ | 92.31 | 86.23 | 98.50 |
| $p = 0.9$ | 24.67 | 16.42 | 99.09 |

**Fine-tuning Details.** On ResNet-18, we fine-tuned the `CIFAR-10`, `CIFAR-100` and ViT-B/16 on `CIFAR-10`, `CIFAR-100` and `ImageNet` datasets. As the datasets vary in size and complexity, we employed a dynamic dropout rate and used a set of different values for the parameter $\alpha$. The exact values of the dropout rate and $\alpha$ were determined based on the specific characteristics of each task. The chosen values for each task and model can be found in Table 5.

**Hyperparameters & Training Setting.** For the fine-tuning process, we used different training hyperparameters for each model. For ResNet-18, we used a batch size of 128 for `CIFAR-10` and `CIFAR-100`. For ViT-B/16, we used a batch size of 32 for `CIFAR-10`, and 256 for `CIFAR-100` and 64 for `ImageNet`. Regarding the training epochs, ResNet-18 was set to 200 for both `CIFAR-10` and `CIFAR-100`. For ViT-B/16, we set the training epochs to 50 for `CIFAR-10` and `CIFAR-100`, and 10 for `ImageNet` in the original ViT-B/16 and our algorithm. Due to the difficulty in achieving convergence with R-Drop, we increased the training epochs to 30 when training `ImageNet`. Furthermore, the image size for ViT-B/16 was set to 384 for `ImageNet`, and 224 for both `CIFAR-10` and `CIFAR-100`.

The exact hyperparameters for each task and model can be found in Table 5. For the `CIFAR-10` and `CIFAR-100` tasks, we employed the SGD optimizer with a learning rate of $1e-2$, while for the `ImageNet` task in ViT-B/16, we utilized the Adam optimizer with a learning rate of $1e-4$. For data augmentation, we exclusively utilized random cropping for the `CIFAR-10` and `CIFAR-100` tasks. However, for the `ImageNet` dataset, we adopted RandAugment (Cubuk et al., 2020).

## C.3 NATURAL LANGUAGE PROCESSING

**Model Details.** Our experiments utilized two pre-trained models: BERT-base and RoBERTa-base.

BERT-base is a transformer-based model with 12 layers, 768 hidden units, and 12 attention heads, totaling 110 million parameters. It was pre-trained on a large corpus of English text from the BooksCorpus ($800M$ words) and English Wikipedia ($2,500M$ words).

RoBERTa-base, on the other hand, is a variant of BERT that uses a larger byte-level BPE vocabulary, longer training time, and different pre-training data. It has the same architecture as BERT-base but was trained on more data ($160GB$ of text).

**Datasets & Evaluation Metrics.** For language understanding tasks, we adhere to the prevalent pre-training and fine-tuning methodology, with the GLUE benchmark serving as the fine-tuning set. Consistent with previous studies, we focus on eight tasks, including single-sentence classification tasks (CoLA, SST-2), sentence-pair classification tasks (MNLI, QNLI, RTE, QQP, MRPC), and the sentence-pair regression task (STS-B). Detailed data statistics can be found in the original paper (Wang et al., 2019). The evaluation metrics for the aforementioned tasks are as follows: The STS-B task is evaluated using the Pearson correlation; The CoLA task is assessed via Matthew's correlation; Both the F-1 score and accuracy are used as metrics for the MRPC and QQP tasks; The remaining tasks (MNLI, QNLI, RTE, SST-2) are evaluated based on accuracy. These metrics collectively provide a holistic assessment of the models' performance across a range of language understanding tasks.

Table 7: Hyperparameters for BERT-base experiments.

| Datasets | $p$ | | | $\alpha$ | | | Learning Rate | Batch Size | Epochs |
|---|---|---|---|---|---|---|---|---|---|
| | BERT-base | + MoReDropL | + MoReDrop | BERT-base | + MoReDropL | + MoReDrop | | | |
| CoLA | 0.1 | 0.7 | 0.2 | - | 1 | 1 | 2.00E-05 | 32 | 3 |
| SST-2 | 0.1 | 0.3 | 0.2 | - | 1 | 1 | 2.00E-05 | 32 | 3 |
| MRPC | 0.1 | 0.9 | 0.3 | - | 2 | 0.1 | 2.00E-05 | 32 | 3 |
| STS-B | 0.1 | 0.3 | 0.1 | - | 0.1 | 1 | 2.00E-05 | 32 | 3 |
| QQP | 0.1 | 0.2 | 0.2 | - | 0.5 | 0.5 | 2.00E-05 | 32 | 3 |
| MNLI | 0.1 | 0.2 | 0.2 | - | 1 | 1 | 2.00E-05 | 32 | 3 |
| QNLI | 0.1 | 0.3 | 0.3 | - | 1 | 1 | 2.00E-05 | 32 | 3 |
| RTE | 0.1 | 0.5 | 0.3 | - | 0.1 | 0.5 | 2.00E-05 | 32 | 3 |

Table 8: Hyperparameters for RoBERTa-base experiments.

| Datasets | $p$ | | | $\alpha$ | | | Learning Rate | Batch Size | Epochs |
|---|---|---|---|---|---|---|---|---|---|
| | RoBERTa-base | + MoReDropL | + MoReDrop | RoBERTa-base | + MoReDropL | + MoReDrop | | | |
| CoLA | 0.1 | 0.5 | 0.2 | - | 0.1 | 1 | 2.00E-05 | 32 | 3 |
| SST-2 | 0.1 | 0.3 | 0.2 | - | 0.1 | 1 | 2.00E-05 | 32 | 3 |
| MRPC | 0.1 | 0.1 | 0.7 | - | 2 | 0.1 | 2.00E-05 | 32 | 3 |
| STS-B | 0.1 | 0.2 | 0.4 | - | 2 | 2 | 2.00E-05 | 32 | 3 |
| QQP | 0.1 | 0.3 | 0.2 | - | 0.5 | 0.5 | 2.00E-05 | 32 | 3 |
| MNLI | 0.1 | 0.3 | 0.2 | - | 0.5 | 1 | 2.00E-05 | 32 | 3 |
| QNLI | 0.1 | 0.1 | 0.2 | - | 0.1 | 1 | 2.00E-05 | 32 | 3 |
| RTE | 0.1 | 0.1 | 0.2 | - | 0.1 | 0.1 | 2.00E-05 | 32 | 3 |

**Baseline Setting & Experiments Design.** Given that both BERT-base and RoBERTa-base incorporate standard dropout, MoReDrop can be directly applied. For MoReDropL, as applied in Appendix C.2, we employed standard dropout before the final linear layer.

**Fine-tuning Details.** Both models were fine-tuned on the GLUE benchmark datasets. As the datasets vary in size and complexity, we employed a dynamic dropout rate and used a set of different values for the parameter $\alpha$. The exact values of the dropout rate and $\alpha$ were determined based on the specific characteristics of each task. The chosen values for each task and model can be found in Table 7 and Table 8.

**Hyperparameters & Training Setting.** For the fine-tuning process, we adhered to the original training hyperparameters used in the BERT and RoBERTa models. For the sake of simplicity in implementation, we assigned the same values for batch size, learning rate, and epochs across different tasks and models, which were 32, $2e-5$, and 3, respectively. We used the Adam optimizer for both models. The precise hyperparameters for each task and model can be found in Table 7 and Table 8.

## C.4 ABLATION STUDY

### C.4.1 SENSITIVITY ANALYSIS

In this section, we conduct an ablation study to evaluate the impact of hyperparameters in MoReDrop, specifically the dropout ratio $p$ and the regularization weight $\alpha$. The sets explored for $p$ and $\alpha$ are $\{0.1, 0.3, 0.5, 0.7\}$ and $\{0.1, 0.5, 1, 2\}$, respectively. As illustrated in Figure 4, MoReDrop exhibits significant robustness to hyperparameter variations. Most combinations yield performance superior to the baseline, with the exception of certain extreme combinations such as $(p, \alpha) = (0.7, 2)$ on the CIFAR-10 dataset using the ResNet-18 backbone, and $(p, \alpha) = (0.7, 1)/(0.5, 2)/(0.5, 1)$ on the MRPC task. Moreover, we observe that the dropout ratio $p$ has a more substantial impact on the

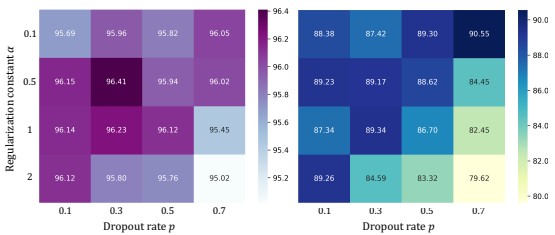

Figure 4: Performance of various tasks using MoReDrop under different hyperparameter combinations. **Left:** Performance of ResNet-18 with DropBlock and MoReDrop on the CIFAR-10 dataset. **Right:** Performance of RoBERTa-base with MoReDrop on the MRPC task.

final performance. Our algorithm demonstrates significant robustness to variations in the regularization weight $\alpha$, attributable to our regularization design which bounds the loss within $(-1, 1)$. Generally, when tuning MoReDrop for application in other domains, we recommend prioritizing adjustments to $p$ due to its significant influence on performance.

Typically, higher dropout rates are associated with less precise predictions. However, we show a superior performance gain of MoReDrop in high dropout rates, even at 0.7, as shown in Figure 4. We here attribute the phenomenon to two key reasons.

The first one is the training paradigm in MoReDrop. In MoReDrop, the primary gradient update mechanism is derived from traditional dense training, distinct from the influence of the regularizer. The regularizer integrates benefits of dropout models into the dense training framework. The parameter $\alpha$ is pivotal in modulating the extent to which the regularizer, denoted as $\mathcal{R}$, impacts the gradient update in the dense model. In scenarios with high dropout rates, a reduced $\alpha$ value minimizes the impact from the regularizer, preserving the performance of the dense model that primarily relies on conventional training methods sans dropout. This approach aligns with findings from our ablation study (Figure 4), indicating the importance of lower $\alpha$ values in sustaining model performance under increased dropout rates. For instance, MoReDrop shows heightened sensitivity to $\alpha$ in a 0.7 dropout scenario in the MRPC task, more so than at lower dropout levels. Additionally, optimal performance at a 0.7 dropout rate necessitates a minimal $\alpha$, specifically around 0.1, to mitigate uncertainties inherent in the dropout model. In contrast, a higher $\alpha$ is advantageous in lower dropout scenarios to borrow advantages from dropout models, leveraging the reliable predictions of the dropout model.

The second one is the bound nature of $\mathcal{R}$. The inherent boundedness of the regularizer $\mathcal{R}$ facilitates constraining the gap to less than 1, coupled with the weight $\alpha$, exerts limited influence on the optimization of the dense model.

### C.4.2 REGULARIZATION LOSS FUNCTION

In prior research, specifically in the contexts of ELD and FD, the $L_2$ distance on hidden states was employed as a regularization loss function. However, this approach diverges significantly from the primary training objective, which is to minimize the negative log-likelihood over the model's output distribution. R-Drop introduced the use of KL-divergence between output probability distributions. While this method imposes a strong constraint on the model, affecting its generalization capabilities, it also incurs considerable computational costs.

To address these challenges, we propose to use a loss scalar to quantify the discrepancy and then project this into a relative space, adjustable through a hyperparameter $\alpha$. Additionally, we exploit the bounded nature of the $sigmoid(\cdot)$ function, within $[0, 1]$, to reshape it into $g(x) = \frac{e^x - 1}{e^x + 1} = 2 \cdot (sigmoid(x) - \frac{1}{2})$, forming our regularization loss function.

We conduct numerical experiments to demonstrate the efficacy of different formulations of $g(\cdot)$ For practicality and simplicity in implementation, we adopt ResNet-18 as our model backbone. We configure four distinct experimental scenarios: ResNet-18-2D, which incorporate 2 dropout layers.

| $g(\cdot)$ | $p = 0.1$ | $p = 0.3$ | $p = 0.5$ | $p = 0.7$ | $p = 0.9$ |
|---|---|---|---|---|---|
| $g(x) = x$ | 95.22 | 95.31 | 95.13 | 95.04 | 90.07 |
| $g(x) = x^2$ | 95.42 | 95.61 | 95.33 | 95.23 | 94.23 |
| $g(x) = \frac{e^x-1}{e^x+1}$ | **95.56** | **95.70** | **95.72** | **95.47** | **95.23** |

Table 9: Results of different form of $g(\cdot)$.

Tables above reveal distinct performance trends across different $g(\cdot)$ functions. Specifically, $g(x) = \frac{e^x-1}{e^x+1}$ outperforms in all settings, $g(x) = x$ demonstrated inferior performance compared to both $g(x) = x^2$ and $g(x) = \frac{e^x-1}{e^x+1}$. Notably, $g(x) = x^2$ and $g(x) = \frac{e^x-1}{e^x+1}$ demonstrate comparable effectiveness under conditions of low dropout rates. However, as dropout rates increase to $0.7, 0.9$, the function $g(x) = \frac{e^x-1}{e^x+1}$ exhibits superior robustness in these extreme conditions. This leads to the conclusion that $g(x) = \frac{e^x-1}{e^x+1}$ is not only adaptable to varying scenarios but also demonstrates enhanced performance and robustness across different dropout rates.

Further investigation focuses on the role of the hyperparameter $\alpha$ within our regularization function, examining whether it is more effective in the form of $\alpha \cdot g(x)$ or $g(\alpha \cdot x)$. To this end, an experiment was conducted using ResNet-18-4D as the backbone, setting $p = 0.5$, and observing performance variations across different values of $\alpha$.

| $g(\cdot)$ | $\alpha = 0.1$ | $\alpha = 0.5$ | $\alpha = 1$ | $\alpha = 2$ | $\alpha = 5$ | $\alpha = 10$ |
|---|---|---|---|---|---|---|
| $\alpha \cdot g(x)$ | **95.55** | 95.81 | 95.61 | **95.48** | 94.28 | 73.80 |
| $g(\alpha \cdot x)$ | 95.52 | **95.83** | **95.66** | **95.48** | **95.36** | **95.22** |

Table 10: Results under different $\alpha$ and $g(\cdot)$.

From the table, we observe that both $\alpha \cdot g(x)$ and $g(\alpha \cdot x)$ deliver comparable performance under low $\alpha$ value. However, with the increment of $\alpha$, $g(\alpha \cdot x)$ performs more robust compared to $g(\alpha \cdot x)$, meanwhile considering various application scenarios, it becomes evident that a bounded loss function like $g(\alpha \cdot x)$ offers more versatility in addressing a range of issues. Therefore, this paper opts for the use of $g(\alpha \cdot x)$ in the proposed framework. Nevertheless, it is crucial to recognize the potential for discovering more effective loss functions. Future research will be directed towards exploring these alternatives to further enhance the performance and adaptability of our model.

