# OpenReview forum: "MoReDrop: Dropout Without Dropping"
_ICLR.cc/2024/Conference — Submitted to ICLR 2024_

### Official Review · Reviewer_8mBK · 2023-10-28

**Soundness:** 2 fair
**Presentation:** 2 fair
**Contribution:** 2 fair
**Rating:** 3
**Confidence:** 3

**Summary:**

The paper unveils MoReDrop and MoReDropL, new strategies crafted to mitigate the challenges of model distributional shifts encountered in dropout models during the training and evaluation phases. MoReDrop emphasizes the dense model loss, which improves the consistency between training and evaluation models but at a substantial computational cost due to added matrix operations across all layers. A lighter version, MoReDropL, addresses these computational concerns, focusing computations on the final layer, albeit at the cost of some level of generalization.

However, the paper falls short in providing exhaustive experimental evidence, leaving some claims, like the mitigation of distributional shifts, unsubstantiated. Additionally, there’s a notable discrepancy in mathematical formulations, particularly in Equation (4), which raises questions about its accuracy and alignment with standard conventions. A lack of thorough clarification regarding the significance of introduced theoretical concepts, such as Theorem 1, leaves the reader ambiguous about their precise role and implications. Moreover, certain aspects, such as the backpropagation process and the methodology’s robustness at varying dropout rates, lack detailed exploration and clarification, making it challenging to grasp the full depth of the proposed strategies' effectiveness and functionality.

**Strengths:**

- **New Approach to Address Distributional Shift:** The paper introduces MoReDrop, an innovative approach in mitigating the model distributional shift encountered during the evaluation phases, which is common with the use of dropout in neural networks.

- **Utilization of Dense Model Loss:** Unlike previous methods that primarily use the sub-model loss, this study emphasizes the use of the dense model loss as the main loss function. This prioritization allows for better consistency between the training and evaluation models.

**Weaknesses:**

- **Lack of Explicit Experimental Results:** The paper claims that MoReDrop mitigates distributional shift issues through a dense-to-sub regularization approach. However, it lacks explicit experimental evidence supporting how this method shows superiority over previous techniques. The absence of clear, comparative results makes it challenging to validate the asserted benefits of MoReDrop.

- **Discrepancy in Mathematical Formulation:** There’s an inconsistency in Equation (4), where the expectation is placed inside the exponent, but during training, the expectation is positioned outside the exponent, leading to potential questions regarding the mathematical precision and validity of the given formulation. This inconsistency needs to be addressed to ensure the robustness and accuracy of the mathematical model presented.

- **Unclear Implications of Theoretical Concepts:** The paper introduces Theorem 1 but fails to elucidate its significance concerning Equation (4). It leaves the readers with ambiguity regarding the theorem's role and impact on the equation's interpretation or formulation.

- **Ambiguity in Backpropagation Process:** The explanation regarding the gradient backpropagation in Section 3.2 is unclear. It doesn’t adequately explain why the shared parameter sub-model $ M_i$ remains unaffected during backpropagation despite its involvement in the equations, leading to confusion about the actual process and mechanisms at play.

- **Questionable Robustness at High Dropout Rates:** The claim regarding MoReDrop’s robustness at high dropout rates appears counterintuitive, and the paper doesn’t provide a comprehensive explanation to dispel this contradiction. It lacks clarity on how MoReDrop maintains effective regularization, especially with high values of $p$, which seems to diminish the effect of $M_i$ and potentially nullify the dropout effect.

- **Lack of Depth in Exploring Dropout Rates:** The paper doesn’t dive deep into exploring the implications of varying dropout rates in MoReDrop. There’s a missed opportunity to clarify the differences and effects at different dropout rates, such as distinguishing between $p=0$ and $p=1.0$, which would have contributed to a richer understanding of the methodology.

For clarifications, please refer to the Questions section.

**Questions:**

-  In the introduction and Section 3.3, the authors assert that MoReDrop alleviates the distributional shift issue by employing a dense-to-sub regularization approach. Could the authors provide clarity on how this method is advantageous over previous strategies? It seems that there might be a lack of explicit experimental results in the paper that substantiate this claim, or perhaps they might have been overlooked.

- Eq (4) in Page 4 incorporates $\mathbb{E}_{S_D}[l(D, S_D;\theta)]$ within the exponent. Given the application of the minibatch and implicit dropout $p$, the expectation operation seems to be applied outside of the exponent. This positioning could be questioned due to $\mathbb{E}\frac{\exp(X)}{\exp(Y)}\neq \frac{\exp(\mathbb{E}X)}{\exp(\mathbb{E}Y)}$ in general,so they are not equivalent. Could the authors elucidate on this inconsistency?

- What is the significance of Theorem 1 in relation to Eq. (4)? Could the authors clarify the implications of the theorem on the formulation or interpretation of this equation?

- Could the authors please clarify the statement in Section 3.2 on Page 4: 'Note that we only apply through gradient backpropagation $M$; shared parameter sub-model $M_i$ does not undergo updates through gradient backpropagation.' From Equation (5), it seems that information from $M_i$ is included, leading me to assume that $M_i\subset M$ would also be influenced during backpropagation. Could you elucidate how $M_i$ remains unaffected?

- In Section 3.3, the authors assert the robustness of MoReDrop at high dropout rates, a claim that appears somewhat counterintuitive. Reference is made to Table 5 on page 13, where MoReDrop exhibits advantages at a high dropout rate, such as $p=0.9$. This, however, prompts a question: with such a high value of $p$, wouldn't the effect of $M_i$ be diminished, causing the regular cross-entropy loss in Equation (5) to predominate, thereby nullifying the dropout effect? Could the authors elucidate how MoReDrop, even at high dropout rates, continues to maintain effective regularization?

- More specifically, what's the difference between $p=0$ and $p=1.0$ in MoReDropout? When $p=0$, what's the implication of the regularization term in Eq. (4)? **Added (as of 11/14)**. When $p=1.0$, what's the value of $\mathbb{E}_{S_D}[l(D, S_D;\theta)]$?

---

> ### Author Response · Authors · 2023-11-15
> **Official Response to Reviewer 8mBK 1/2**
>
> > W1 and Q1: Lack of Explicit Experimental Results: The paper claims that MoReDrop mitigates distributional shift issues through a dense-to-sub regularization approach. However, it lacks explicit experimental evidence supporting how this method shows superiority over previous techniques. The absence of clear, comparative results makes it challenging to validate the asserted benefits of MoReDrop.
>
> Answer: We thank the reviewer for raising this important question and we have revised this paper for further clarification. Aside of the surprising performance on the benchmark, we would like to clarify that **the model distributional shift is 0** in our proposed method without explicit experimental evidence.
>
> In response, we clarify that the key contributor to the model distributional shift is the active updating of dropout (sub-) models. In other words, the gap exists and can be represented by $\mathcal{G}$ as long as the dropout model performs the gradient backpropagation.
>
> However, MoReDrop actively updates only the dense model; the sub-model is passively updated due to shared attributes. This maintains the model consistency, i.e., the dense model, during training and evaluation, as so the distributional shift is zero. It has no relationship between $\mathcal{G}$ because we never perform gradient backpropagation on the shared-parameter sub-model. Besides, the regularizer retains dropout benefits in the dense model. We also present a clear comparison between MoReDrop and previous methods in Appendix Table 4:
>
> | Algorithms | Gradient back-propagation | Dense model update | Sub-model update | Regularizer | Distribution shift |
> |------------|---------------------------|--------------------|------------------|-------------|-------------------|
> | FD [[Zolna et al., 2018](https://arxiv.org/pdf/1711.00066.pdf)] | Sub-model | Passive | Active | Sub-to-sub | $\mathcal{G}$ |
> | R-Drop [[Liang et al., 2021](https://proceedings.neurips.cc/paper/2021/file/5a66b9200f29ac3fa0ae244cc2a51b39-Paper.pdf)] | Sub-model | Passive | Active | Sub-to-sub | $\mathcal{G}$ |
> | WordReg [[Xia et al., 2023](https://ieeexplore.ieee.org/stamp/stamp.jsp?tp=&arnumber=10095552)] | Sub-model | Passive | Active | Sub-to-sub | $\mathcal{G}$ |
> | ELD [[Ma et al., 2017](https://arxiv.org/pdf/1609.08017.pdf)] | Sub-model | Passive | Active | Dense-to-sub | $\mathcal{G}$ |
> | MoReDrop (ours) | Dense model | Active | Passive | Dense-to-sub | 0 |
>
> > W2 and Q2: Discrepancy in Mathematical Formulation: There’s an inconsistency in Equation (4).
>
> Answer: Thank the reviewer for pointing out the potential misunderstanding regarding Equation (4) in our manuscript. We would like to clarify there is no inconsistency in Eq (4). Concretely, this expectation over $S_D$ in $\mathbb{E}_{S\_{\mathcal{D}}}\left[l\left(\mathcal{D}, S\_{\mathcal{D}}; \theta\right)\right]$ is designed to approximate dropout as an ensemble of neural networks via weight sharing during the training phase, as defined in Eq (1). However, it does this **without an explicit expectation operator** over $S_D$.
>
> In contrast, during the training process with mini-batches, the expectation is taken over the data instead. To aid in understanding and implementing our approach, we have included a clear pseudocode formatted in a Pytorch-like style, in Appendix A under Algorithm 2. This should provide further clarification and assistance in grasping the nuances of our method.
>
> > W3 and Q3: Unclear Implications of Theoretical Concepts.
>
> Answer: We would like to clarify that Theorem 3.1 is fundamental in establishing the validity and effectiveness of **the first-order form**, i.e., $L_1$, of our loss function with the non-decreasing function $g$:
> $$
> \left.\mathcal{R}=g\left(\mathbb{E}_{S\_{\mathcal{D}}}\left[l\left(\mathcal{D}, S\_{\mathcal{D}} ; \theta\right)\right]-l(\mathcal{D} ; \theta)\right]\right).
> $$
>
> Theorem 3.1 states that $l(\mathcal{D} ; \theta) \leq \mathbb{E}_{S\_{\mathcal{D}}}\left[l\left(\mathcal{D}, S\_{\mathcal{D}} ; \theta\right)\right]$.
>
> This inequality ensures that the expression $\mathbb{E}_{S\_{\mathcal{D}}}\left[l\left(\mathcal{D}, S\_{\mathcal{D}}; \theta\right)\right]-l(\mathcal{D}; \theta)$ is always greater than or equal to 0.
>
> Without Theorem 3.1, if we suppose $\mathbb{E}_{S\_{\mathcal{D}}}\left[l\left(\mathcal{D}, S\_{\mathcal{D}}; \theta\right)\right]$ is smaller than $l(\mathcal{D}; \theta)$, we would face the paradoxical situation to minimize the loss function, i.e., increasing  $l(\mathcal{D}; \theta)$, that does not align with our primary goal. Then, we have to choose the second order form, e.g., $L_2$,  to construct the regularizer.
>
> In summary, Theorem 3.1 is important which guarantees $x\geq 0$  and thus $\min_x \mathcal{R}= g(0)$ if $g$ is a non-decreasing function. We then can simply set the $g$ as the $L_1$ norm rather than the $L_2$ norm. The ablation study for $g$ in Appendix C.4.2 demonstrates the efficiency of $L_1$ choice, especially in high dropout rates.

---

> ### Author Response · Authors · 2023-11-15
> **Official Response to Reviewer 8mBK 2/2**
>
> > W4 and Q4: Ambiguity in Backpropagation Process: The explanation regarding the gradient backpropagation in Section 3.2 is unclear. It doesn't adequately explain why the shared parameter sub-model $M_i$ remains unaffected during backpropagation despite its involvement in the equations, leading to confusion about the actual process and mechanisms at play.
>
> Answer: We apologize for any misinterpretation of the claim that "the shared-parameter sub-model $M_i$ remains unaffected during backpropagation", and we have revised our paper for clarification. In response, this claim **does not exist in our manuscript**. Actually, the parameter of $M_i$ is passively updated through the shared nature with the dense model with active updates. To aid in understanding and implementing our approach, we have included a clear pseudocode formatted in a Pytorch-like style, in Appendix A under Algorithm 2. This should provide further clarification and assistance in grasping the nuances of our method.
>
> > W5: Questionable Robustness at High Dropout Rates: The claim regarding MoReDrop's robustness at high dropout rates appears counterintuitive, and the paper doesn't provide a comprehensive explanation to dispel this contradiction. It lacks clarity on how MoReDrop maintains effective regularization, especially with high values of $p$, which seems to diminish the effect of $M_i$ and potentially nullify the dropout effect.
>
> Answer: We thank the reviewer for pointing out this important question and we have revised our paper for better clarification. We here clarify for two main reasons:
>
> **The training paradigm of MoReDrop**. In MoReDrop, the primary gradient update mechanism is derived from traditional dense training, distinct from the influence of the regularizer. The regularizer integrates the benefits of dropout models into the dense training framework. The parameter $\alpha$ is pivotal in modulating the extent to which the regularizer, denoted as $\mathcal{R}$, impacts the gradient update in the dense model. In scenarios with high dropout rates, a reduced $\alpha$ value minimizes the impact from the regularizer, preserving the performance of the dense model that primarily relies on conventional training methods sans dropout.
>
> This approach aligns with findings from our ablation study in Appendix C.4.1, indicating the importance of lower $\alpha$ values in sustaining model performance under increased dropout rates. In contrast, a higher $\alpha$ is advantageous in lower dropout scenarios to borrow advantages from dropout models, leveraging the reliable predictions of the dropout model.
>
> **The bound nature of $\mathcal{R}$.** The inherent boundedness of the regularizer $\mathcal{R}$ facilitates constraining the gap to less than 1, coupled with the weight $\alpha$, exerts limited influence on the optimization of the dense model.
>
> In summary, in high dropout rates, the performance is retained mostly from the dense model by setting a smaller $\alpha$, this inherits from the unique training paradigm of MoReDrop and the bound nature of the regularizer.
>
> > W6 and Q6: Lack of Depth in Exploring Dropout Rates: The paper doesn't dive deep into exploring the implications of varying dropout rates in MoReDrop. There's a missed opportunity to clarify the differences and effects at different dropout rates, such as distinguishing between $p=0$ and $p=1.0$, which would have contributed to a richer understanding of the methodology.
>
> Answer: I would like to clarify that we have performed a comprehensive dropout rate set, i.e., [0.1, 0.2, 0.3, 0.4, 0.5, 0.7, 0.9] in our experiments. Furthermore, we believe it is not entirely logical to conduct the extreme dropout cases.
> 1. $p=0$: $p=0$ indicates that there is no dropout on the sub-model, which means the dense model and the sub-model are identical with no chance to leverage the advantages of dropout models. This also means that the regularization term $\mathcal{R}$ in Eq. 4 is 0 because $\mathbb{E}_{S\_{\mathcal{D}}}\left[l\left(\mathcal{D}, S\_{\mathcal{D}} ; \theta\right)\right]=l(\mathcal{D} ; \theta)]$.
> 2. $p=1$: $p=1$ indicates that the sub-model is nothing because we mask all neurons, which means the sub-model does not exist and the prediction of it is totally random, where we do not expect any information or benefits from it. The value of $\mathbf{\hat{y}=f(\mathcal{D}; \theta)}$ will be a random prediction without any improvement. Practically, in a dropout setting with  $p=1$ **in layer $l$**, the output of the **layer $l$** will invariably be zero.

---

> ### Comment · Reviewer_8mBK · 2023-11-16
>
> Thank you for the quick response. I may have additional questions later for other issues, but before we delve deeper, I'd like to ask a quick question for clarification. I remain unclear about the scenario when $p=1.0$ case (or when $p\to 1.0$).
>
> When $p=1.0$, I think it's reasonable to assume that $\mathbb{E}_{S_D}[l(D, S_D;\theta)]=0$ because there's nothing left in $S_D$. i.e., $S_D=\emptyset$. Then by Theorem 3.1, $l(\mathcal{D},\theta)=0$. Could the authors please provide some clarification on how to interpret this phenomenon?
>
> Or what's the meaning that $\mathbb{E}_{S_D}[l(D, S_D;\theta)]$ will be a random value when $p=1.0$? How is it possible that the expectation value could be a random number? Thank you very much.

---

> ### Author Response · Authors · 2023-11-16
> **Response to the Question Regarding p=1**
>
> Thanks for the further question! We find that we have made an important misunderstanding: We refer to the output prediction as $\hat{y}=f(D; \theta)$ rather than the loss function $\mathbb{E}_{S\_{\mathcal{D}}}\left[l\left(\mathcal{D}, S\_{\mathcal{D}}; \theta\right)\right]$ from the sub-model with dropout. We have also clarified our last response in bold.
>
> We would like to clarify that **every dropout model is an implicit ensemble for many models with shared parameters**, so you can safely ignore it.
>
> In each training step, a layer $l$ with dropout $p=1$, see the [PyTorch source code](https://github.com/pytorch/pytorch/blob/main/torch/_decomp/decompositions.py#L1097), will output a matrix equivalent to zeros. This behavior effectively blocks the propagation of input information from $\mathcal{D}$ through the network beyond this layer. Consequently, the final output of the neural network $\hat{y}=f(\mathcal{D} ; \theta)$ is transformed into $\hat{y}=f^{\prime}(\mathbf{O}; \theta')$, where $f^{\prime}(\cdot;\theta')$ is the subsequent layers after the dropout layer up to the output and $\mathbf{O}$ is a zero-like matrix. This results in the output prediction $\hat{y}$ being effectively random and devoid of meaningful information derived from the input $\mathcal{D}$ with $p=1$. However, we would like to highlight that $p=1$ is not entirely logical. Here we give a simple example:
> ```python
> import torch
> from torch.nn import Dropout
>
> input = torch.tensor([0,1,2,3])
>
> drop_0 = Dropout(0) # Instantiate a dropout function with p=0
> drop_1 = Dropout(1) # Instantiate a dropout function with p=1
>
> output_0 = drop_0(input) # tensor([0,1,2,3])
> output_1 = drop_1(input) # tensor([0,0,0,0])
> ```
>
> If you have any additional questions or need further clarification, please let us know. Once all your queries have been satisfactorily addressed, we kindly ask you to consider raising your rating.

---

> > ### Comment · Reviewer_8mBK · 2023-11-16
> >
> > I appreciate the authors responding promptly once again. I was asking for $p=0$ and $p=1.0$ cases to check the mathematical consistency of the proposed framework. It seems that the paper couldn't solve the consistency issue.
> >
> > Moreover, since the expectation is applied **throughout the whole training time**, as mentioned in the above response, we can go back to my 2nd question regarding the position of the expectation in the equation. If so, it's reasonable to think the expectation operator is outside the regularization term. i.e. $\mathcal{R} = \mathbb{E}_{S_D}g([l(D, S_D; \theta)] − l(D; \theta)])$. Then, the entire argument cannot be valid after that.
> >
> > Therefore, I keep my score as-is unless I have a severe misunderstanding about this issue.

---

> ### Author Response · Authors · 2023-11-16
> **Further Response to the Consistency Consideration of Eq (4)**
>
> We thank the reviewer for pointing out this important consideration. In response, we acknowledge that our last response is somehow wrong and needs further clarification. However, we would like to clarify with that Eq (4) in training is consistent.
>
> 1. We clarify that dropout aims at, **simultaneously and jointly**, training an ensemble of exponentially many neural networks (one for each conﬁguration of dropped units) while sharing the same weights/parameters. (see almost the same context in [Liang et al. (2021)](http://arxiv.org/abs/2106.14448) (page 1, Section 1)  [Ma et al., 2017](https://openreview.net/forum?id=rkGabzZgl) (page 2, Section 2.2)). In other words, despite the presence of only a single explicit dropout model during training at every training step, this effectively represents implicitly an ensemble of numerous dropout models due to the variability introduced by the set of dropout random variables $S$. More evidence can also be found in Eq (2) in [Liang et al. (2021)](http://arxiv.org/abs/2106.14448).
> 2. Further, we have also conducted explicit experiments showing the consistency:  $ \mathbb{E}\_{S\_\mathcal{D}}\left[l\left(\mathcal{D}, S\_{\mathcal{D}}; \theta\right)\right] \geq l( \mathcal{D}; \theta)$. We would be happy to provide more experiment clarification or details if the reviewer 8mBK needs them.
>
> In conclusion, Eq (4) exhibits consistency because **every dropout model is an implicit ensemble for many models with shared parameters**, so you can safely ignore it. We also want to share our gratitude to reviewer 8mBK for discussing this with us and we will highlight it in our further manuscript. Additionally, we have amended our previous unclear response with bold.

---

> ### Comment · Reviewer_8mBK · 2023-11-18
>
> Thank you once again for your prompt response.
>
> - Then, it clarifies that the consistency question can be resolved. Ma et al. (2017) suggest that $\mathbb{E}_{S_D}[l(D, S_D;\theta)]$ tends towards infinity as $p \to 1.0$, due to $p(y_i|x_i, S_i, \theta)$ approaching zero in the same limit. Given this, I observe no substantial distinction between Theorem 1 in Ma et al. (2017) and Theorem 1 in the current paper. Consequently, **it would be appropriate for the authors to reference Ma et al. (2017) again in your Theorem 1, acknowledging the same findings.**
> - As $p \to 1$, the regularization term $\mathcal{R}$ converges to 1, indicating a diminishing impact on regularization. Consequently, this suggests **the necessity for extended training duration to achieve a suitable stationary state in the network for high $p$**. Otherwise, there's no point in regularization.
> - So, the paper assumes log-likelihood loss. Can we extend this framework to **cross-entropy loss**?
> - Most importantly, the training-time ensemble of sub-networks suggests that the expectation should be placed **outside the function $g(\cdot)$, leading to $\mathcal{R} = \mathbb{E}g([l(D, S_D; \theta)] - l(D; \theta))$**. As a result, **Theorem 1 and subsequent analyses cannot be valid** as pointed out above + my 2nd question. i.e. the paper overlooks a crucial proof step for $\mathcal{R} = \mathbb{E}g([l(D, S_D; \theta)] - l(D; \theta)) = g([\mathbb{E}l(D, S_D; \theta)] - l(D; \theta))$ to make the whole discussion valid. However, I don't think this equality is generally true even when $g$ is non-decreasing.
>
> Please note that my final bullet point raises the most critical question. I expect further clarification on this matter. Thank you.

---

> > ### Author Response · Authors · 2023-11-19
> > **Official Response to Reviewer 8mBK**
> >
> > >1. The same finding with Ma et al. (2017).
> >
> > We appreciate the reviewer for highlighting this point, and we will certainly revise it in our future manuscript. Nonetheless, we would like to emphasize that we have correctly applied Theorem 1 in constructing our regularizer using the $L_1$ distance.
> >
> > > when $p \rightarrow 1$, it is necessity for extend training duration to achieve a suitable stationary state in the network for high $p$.
> >
> > In response, we would like to highlight again the training paradigm of MoReDrop with two steps: (a). the primary loss optimization only on the dense model, ensuring the consistency of the model for training and testing. (b). The regularizer retains dropout benefits, e.g., generalization. In the extreme case, i.e., $p \rightarrow 1$, the performance of the dropout model can be hardly be reliable, thus we essentially need to decrease the $\alpha$ to reduce the influence of the regularizer to gradient update. We present our ablation study of $(p, \alpha)$ for further clarification in Fig 4 of Appendix C.4.1:
> >
> > | (p=0.7, $\alpha$) | Baseline | 0.1   | 0.3   | 0.5   | 0.7   |
> > |-------------------|----------|-------|-------|-------|-------|
> > | CIFAR-10          | 95.53    | 96.05 | 96.02 | 95.45 | 95.02 |
> > | MRPC              | 87.50    | 90.55 | 84.45 | 82.45 | 79.62 |
> >
> > In the table above, even in $p=0.7$ that does not approximate $p\rightarrow 1$, higher $\alpha$ will notoriously undermine the performance, even inferior to the baseline performance, $p=(0.5, 0.7)$  in the CIFAR-10 task and the $p=(0.3, 0.5, 0.7)$. Further, in our suggestion for hyper-parameters search, we claim in our paper "Generally, when tuning MoReDrop for application in other domains, we recommend prioritizing adjustments to $p$ due to its significant influence on performance." (page 16). Based on this fact, we assume it is not entirely logical to use $p \rightarrow 1$ in the practical training under the same training time. Regarding the extended training time for $p \rightarrow 1$, it is not reasonable to opt for an inferior hyper-parameter (given all other settings, i.e., training time, remains constant) due to its hypothetical performance improvement with increased training time.
> >
> > > Extend to cross-entropy loss
> >
> > Maximum log-likelihood estimation is a general idea in statistics. In our experiments with specific tasks, we use the specific extended form of log-likelihood, i.e., cross-entropy in class classification (except the STS-B task) and MSE in the regression task (STS-B).
> >
> >
> > > The crucial proof step regarding the expectation should be outside or inside of $g$ in practical training.
> >
> > We apologize for our previous erroneous conclusions statement "the training-time ensemble of sub-networks" and we have also revised it. In response, our answer regarding this is the same as our last response: every dropout model is an implicit ensemble for many models with shared parameters, so you can safely ignore it. If not in this case with an implicit ensemble, Eq (3) in our paper and Eq (2) in Ma et al. (2017) should not involve the expectation term.
> >
> > Regarding the explicit ensemble operator over the training time, our response is: yes, it has, but the two expectation operators are totally different. As a result, Eq (4) is valid as it only cares about the implicit one.

---

> > > ### Comment · Reviewer_8mBK · 2023-11-21
> > >
> > > I'm very grateful for the authors' engagement in this discussion. The information provided has been sufficient for finalizing my score. Should I have further questions, I'll add more comments.

---

> > > > ### Author Response · Authors · 2023-11-21
> > > > **Consider Raising the Score**
> > > >
> > > > Dear Reviewer, thank you for your valuable feedback; if we have addressed your concerns with the revised paper and response, we kindly ask for a reevaluation of your rating. We're committed to addressing any additional concerns you may have.

---

> ### Comment · Reviewer_8mBK · 2023-11-21
>
> Thank you very much. I maintain my score as before because
> - The authors mainly claim that the distributional shift $\mathcal{G} = 0$ shown in Table 4 because the dense model has been involved during the training time. But this conclusion has a serious logical jump. After the training, the final network we obtain would be **the mixture of the dense network and the stochastic subnetwork**. This means the final network is an expectation of the dense+sub networks throughout the whole training step. Then, the expectation operator still plays. Therefore, the proposed framework cannot remove the distributional shift gap. So $\mathcal{G} = 0$ is a false claim. **If not, please consider providing detailed proof of why the gap must be zero.** If so, it would be a highly non-trivial property.
>    - If we follow the same argument from the authors, the simplest way to remove the distributional shift is without applying dropouts throughout the whole training step. Then, there would be no stochasticity within the network.
> - Because of the above reasons, we **cannot safely ignore** the expectation outside the regularization term $\mathcal{R}$. Therefore, the regularization term still needs to keep the expectation operator. If not, please also consider providing detailed proof of how we can safely ignore the case.

---

> > ### Author Response · Authors · 2023-11-21
> > **Official Response to Reviewer 8mBK**
> >
> > > The final network we obtain would be the mixture of the dense network and the stochastic subnetwork.
> >
> > In response, we clarify that the final network is not a mixture of the dense network and the stochastic subnetwork w/ dropout and $\mathcal{G}=0$. The fundamental reason for this clarification is that MoReDrop never actively updates the shared-parameter sub-model with dropout, which is also a significant ingredient of our proposed method: we only actively update the dense model, and the sub-models are passively updated through parameter sharing with the dense model in each iteration.
> >
> > In other words, MoReDrop always maintains the same model, i.e., dense model, for training and evaluation to ensure $\mathcal{G}=0$. The benefits from the sub-models are preserved through the regularizer.

---

> > > ### Comment · Reviewer_8mBK · 2023-11-21
> > >
> > > Thank you for the clarification again. Please formulate and prove the argument **"we only actively update the dense model, and the sub-models are passively updated through parameter sharing with the dense model in each iteration"** in a rigorous form. I suggest formulating the gradient decomposition of dense and sub-networks and then demonstrating how the regularization term cannot affect the sub-model.

---

> > > > ### Author Response · Authors · 2023-11-22
> > > > **Official Response to Reviewer 8mBK**
> > > >
> > > > Thanks for your query about the gradient decomposition of the updating process. We provide the gradient update of the dense model in the following for the dense model:
> > > > \begin{aligned}
> > > > \nabla_{\theta}f=&\nabla_{\theta}(-\log p(y \mid x ; \theta)+g(\mathbb{E}\_{S\_D} \log p(y \mid x, S_D ; \theta)-\log p(y \mid x ; \theta))) \newline
> > > > =&\nabla_{\theta}(-\log p(y \mid x ; \theta)+\nabla_{\theta}(g(k-\log p(y \mid x ; \theta))) \newline
> > > > =&-\nabla_{\theta}\log p(y \mid x ; \theta) - g\prime(k-\log p(y \mid x ; \theta))\nabla_{\theta} \log p(y \mid x ; \theta)
> > > > \end{aligned}
> > > >
> > > > In the second line, $\mathbb{E}\_{S\_D} \log p(y \mid x, S_D ; \theta)$ equals a constant, e.g., $k$, becuase we stop gradient in the sub-model. Because we **NEVER** update the sub-model, so there is no gradient decomposition of the sub-model.
> > > >
> > > > > Prove the argument "We only actively update the dense model, and the sub-models are passively updated through parameter sharing with the dense model in each iteration"
> > > >
> > > > Given that our algorithm naturally demonstrates this concept in our algorithm, making a proof for it seems not so logical.
> > > >
> > > > > The regularization term cannot affect the sub-model.
> > > >
> > > > We claim that the regularization term **DOES** influence the sub-model, as it shares the entire network with the dense model, except for the dropout variables.

---

> ### Comment · Reviewer_8mBK · 2023-11-22
>
> Dear Authors,
>
> Thank you once again for your engagement in this discussion. As the discussion period nears its conclusion, I plan to elevate these points in the reviewer's discussion. For a better decision, please consider clarifying the following questions.
>
> - I could not find **any stop gradient operation** for the sub-model in Appendix A's Algorithm 2 (Pseudocode). Isn't it contradicting the stop gradient explanation above? Or do you **copy a sampling value** of $l(D, S_D, \theta)$ without a gradient after a single dropout?
> - According to proofs of Theorem 1 from both the paper and Ma et al., 2017, all assume that $\mathbb{E}_{S_D}l(D, S_D, \theta)$ is the expected value across the whole training step. But, the gradient form above says **the expectation has been applied to a single dropout sampling** for each iteration. How is it possible to apply an expectation operator for this single dropout? The proof, the paper's algorithm, and the explanations are inconsistent.
> - The negative log-likelihood losses such as Eq (2) and Eq(3) do not involve any ground truth label information. But Pseudocode in Algorithm 2 uses cross-entropy loss, which involves the ground truth. How can we understand this discrepancy? Note that Eq(2) and Eq(3) are NOT the same formulae of NLLLOSS in Pytorch: https://pytorch.org/docs/stable/generated/torch.nn.NLLLoss.html. So, we cannot naturally extend Eq(2) and Eq(3) to cross-entropy loss.
> - Even if we remove the gradient information from $k$, the above $k$ and $g(k-\log p(\cdot))$ are still random variables throughout the whole training step. Therefore, **the proposed algorithm optimizes a completely different objective function** $-\log p(y|x,\theta)+\mathbb{E}_{k}g(k-\log p(y|x,\theta))$. Please note that the expectation is still placed outside of g.

---

> ### Author Response · Authors · 2023-11-22
>
> We thank the reviewer for joining our further discussion. Here we provide our responses one by one:
>
> > stop gradient operation for the sub-model in Appendix A's Algorithm 2 (Pseudocode).
>
> We do not implement an explicit stop gradient operator in Algorithm 2 but it does only update the dense model. As you can see in Algorithm 2, we first call the dropout model in Lines 8-9, and call the dense model in Lines 12-13. So the optimizer  `optimizer = torch.optim.SGD(model. parameters())` and `loss.backward()` only update the dense model, where the `model. parameters()` is from the dense model without dropout masks. That is "we only actively update the dense model, and the sub-models are passively updated through parameter sharing with the dense model in each iteration" as our previous response.
>
> On the other hand, if we call the dense model first and then call the sub-model, `model. parameters()` is then from the sub-model with dropout masks. As a result, in this case, we update the sub-model. The order of these two models is important as to which model is updated.
>
> > According to proofs of Theorem 1 from both the paper and Ma et al., 2017, all assume that $\mathbb{E}_{S_D} l\left(D, S_D, \theta\right)$ is the expected value across the whole training step. However, the gradient form above says the expectation has been applied to a single dropout sampling for each iteration. How is it possible to apply an expectation operator for this single dropout? The proof, the paper's algorithm, and the explanations are inconsistent.
>
> We have clarified before and we would like to highlight again: A single explicit dropout model during training at every training step represents **implicitly an ensemble of numerous dropout** models due to the variability introduced by the set of dropout random variables $S$. This implicit expectation operators at the dropout random variables $S$, so we can safely ignore it. Ma et al., 2017 also highlight **simultaneously and jointly** to align our clarification.
>
> Regarding the explicit ensemble operator over the training time, as you referring, is an explicit ensemble of different dropout models $p$, rather than the dropout random variables $S$. We can also say it is an expectation over the $S$ but we say this way to avoid confusing those two. These two expectations exhibit fundamental discrepancies. As a result, Eq (4) is valid as it only cares about the implicit one. In our derivation, for simplicity, we omit the expectation over the dataset and dropout models $p$.
>
>
> > The negative log-likelihood losses such as Eq (2) and Eq(3) do not involve any ground truth label information. But Pseudocode in Algorithm 2 uses cross-entropy loss, which involves the ground truth. How can we understand this discrepancy? Note that Eq(2) and Eq(3) are NOT the same formulae of NLLLOSS in Pytorch: https://pytorch.org/docs/stable/generated/torch.nn.NLLLoss.html. So, we cannot naturally extend Eq(2) and Eq(3) to cross-entropy loss.
>
> Eq (2) and Eq(3) do involve the ground truth label information in $\mathcal{D}=\{(x\_1, y\_1), \ldots,(x\_N, y\_N)\}$, as we mentioned in the preliminary.  Furthermore, This approach is fundamentally aligned with the definition of cross-entropy loss, particularly when incorporating a softmax layer prior to the computation of NLL. As noted in the [PyTorch cross-entropy source code](https://pytorch.org/docs/stable/generated/torch.nn.CrossEntropyLoss.html#torch.nn.CrossEntropyLoss): "Note that this case is equivalent to applying LogSoftmax on an input, followed by NLLLoss."
>
>
> > Even if we remove the gradient information from $k$, the above $k$ and $g(k-\log p(\cdot))$ are still random variables throughout the whole training step. Therefore, the proposed algorithm optimizes a completely different objective function $-\log p(y \mid x, \theta)+\mathbb{E}_k g(k-\log p(y \mid x, \theta))$. Please note that the expectation is still placed outside of g.
>
> We think it should optimize a different objective function compared to the vanilla dense model optimization, but all related to the dense model, which is one of the key ingredients of performance improvement in MoReDrop. Regarding the expectation term, we have explained in your question "According to proofs of Theorem 1 from both the paper and Ma et al., 2017 ".

---

### Official Review · Reviewer_57g8 · 2023-10-30

**Soundness:** 3 good
**Presentation:** 3 good
**Contribution:** 3 good
**Rating:** 6
**Confidence:** 2

**Summary:**

The study introduces "MoReDrop," a method designed to address distributional shifts in dropout models without the need for dropping neurons during gradient backpropagation. This approach uses a primary loss function derived from the dense model and incorporates a model gap approximation between the dense model and a sampled sub-model. Experimental results indicate superior performance for MoReDrop and its lightweight version, MoReDropL, across various tasks, with MoReDropL notably excelling in the RTE task from the GLUE benchmark.

**Strengths:**

**Originality**: The paper showcases a novel approach, MoReDrop, which uniquely addresses the distributional shifts in dropout models without resorting to the conventional method of dropping neurons during gradient backpropagation. This inventive technique, particularly the integration of the model gap approximation, distinguishes it from previous studies, adding a layer of originality.

**Quality**: The research provides robust experimental results, indicating the effectiveness of both MoReDrop and MoReDropL across a range of tasks. Moreover, the performance of MoReDropL, especially in the RTE task from the GLUE benchmark, highlights the quality and potential of the proposed methods.

**Clarity**: The paper is well-structured and articulates the core concepts, methodologies, and findings in a comprehensible manner. The delineation between MoReDrop and its lightweight counterpart, MoReDropL, is clear, aiding readers in understanding the nuances and applications of each.

**Significance**: The study's contribution, particularly in the realm of managing model distributional shifts within dropout models, holds significant implications for the broader neural network community.

**Weaknesses:**

The paper does not present glaring weaknesses. However:

1) **Performance and efficiency**: The performance improvement showcased in the experiments is marginal. Across various datasets, there's only about a 1% enhancement. Given this, the results might fall within the error bar, which the authors should consider highlighting. Taking into account the longer training time that MoReDrop demands, the trade-off between model complexity and experimental outcomes seems limited.


2) **Generalizability and Scalability**: While the paper mentions potential scalability issues in challenging domains such as self-supervised learning and reinforcement learning, it would be beneficial for the authors to delve deeper into these concerns. Understanding how MoReDrop would fare in these more complex scenarios, or providing preliminary tests, would strengthen the paper's comprehensiveness.

3) **Comparison with Other Methods**: The paper could benefit from a more extensive comparison with existing methods or techniques that also aim to address distributional shifts in dropout models. By directly contrasting MoReDrop's performance, advantages, and limitations with other prevalent methods, readers would gain a clearer understanding of its position in the current landscape of neural network regularizers.

**Questions:**

1. **Effect of Increasing the Number of "M" Models**:
   - **Question**: How does the performance of MoReDrop change when the number of "M" models is increased? Does the method scale well with an increased number of models, or is there a saturation point beyond which performance gains are minimal or even negative? The broader a model is, the more the model space it can potentially explore. How does MoReDrop perform when applied to broader models? Is there a significant difference in performance compared to narrower models? As the model becomes wider and potentially explores more model spaces, how does this impact the training time, especially when using MoReDrop? Is there an exponential increase, or does the method manage to keep the training time within reasonable bounds?

---

> ### Author Response · Authors · 2023-11-15
> **Official Response to Reviewer 57g8**
>
> > W1: Performance and efficiency: The performance improvement showcased in the experiments is marginal. Across various datasets, there's only about a 1% enhancement.
>
> Answer: We thank the reviewer for suggestion on our experiments and we have added error bars and standard variance calculations to our experimental results under 5 different seeds in our new manuscript.  While we acknowledge that a 1% improvement might initially appear marginal, it is important to consider the context within the field of computer vision, where even small enhancements can be significant, especially when working with high-performing baseline models.
>
> > W1: Performance and efficiency: Taking into account the longer training time that MoReDrop demands, the trade-off between model complexity and experimental outcomes seems limited.
>
> Answer: We thank the reviewer for the attention on the efficiency of MoReDrop. We would like to clarify that we have consider this and we propose MoReDropL, a light version of MoReDrop.  MoReDropL maintains the essence of the original approach while significantly reducing the computational overhead. In our experiments, MoReDrop achieves superior performance compared to the baseline models in many tasks (Table 1, 2), yet is nearly as efficient as the baselines (Table 3).
>
> > W2: Generalizability and Scalability: While the paper mentions potential scalability issues in challenging domains such as self-supervised learning and reinforcement learning, it would be beneficial for the authors to delve deeper into these concerns. Understanding how MoReDrop would fare in these more complex scenarios, or providing preliminary tests, would strengthen the paper's comprehensiveness.
>
> Answer: Thank the reviewer for highlighting this important question. Challenging tasks exhibit unique issues inherent to their respective domains. For instance, in reinforcement learning, the non-stationarity of the learning target, stemming from bootstrapping and data flow, necessitates specialized designs to address these concerns in addition to the model distributional shift. Exploring the application of MoReDrop in these challenging domains constitutes a key direction for our future research.
>
> > W3: Comparison with Other Methods.
>
> Answer: Thank you for your feedback on our manuscript on the comparison with other methods. We appreciate the opportunity to clarify key aspects of our work. In our study, we have conducted thorough experiments to demonstrate MoReDrop's effectiveness, covering a wide range of tasks from image processing to language tasks, as detailed in Sections 4.1 and 4.2. Additionally, we provide a comprehensive loss analysis in Section 4.3 to illustrate MoReDrop's superiority over R-Drop.
>
> We also included detailed sections on training methodologies (Section 4.4), a hyper-parameter sensitivity analysis (Appendix C.4.1), and an ablation study focusing on the loss function (Appendix C.4.2). These sections further establish the robustness and effectiveness of MoReDrop.
>
> Regarding the selection of baselines, we acknowledge the limitations of choice in model distributional shifts within dropout methods. Our choice of state-of-the-art baselines, particularly R-Drop, and various dropout methods, was strategic to showcase MoReDrop's scalability and applicability across different dropout techniques. This approach ensures a comprehensive evaluation of MoReDrop's performance in contrast to existing methods.
>
> > Q1: Effect of Increasing the Number of "M" Models
>
> Answer: Thank you for bringing this to our attention. To clarify, the focus of our paper
> focus on the model distribution shift between the main model and its shared-parameter sub-models, and does not extend to the specific number of "M" models used. It's commonly understood that increasing the number of models can lead to performance improvements, but this also comes with significant increases in computational and memory costs.

---

> > ### Comment · Reviewer_57g8 · 2023-11-18
> > **Thank you for the response**
> >
> > Thank you very much for your detailed response. It has successfully addressed most of my concerns and questions.
> > Based on the comprehensive reply, I keep my score and lean towards accepting the paper.

---

### Official Review · Reviewer_2qPD · 2023-10-31

**Soundness:** 3 good
**Presentation:** 3 good
**Contribution:** 2 fair
**Rating:** 5
**Confidence:** 4

**Summary:**

This paper introduces an approach for training a neural network, involving regularization with a separate network that incorporates dropout. The paper is well-structured and fairly easy to follow.

**Strengths:**

- The idea is simple and easy to implement

**Weaknesses:**

1. Were there any experiments conducted using a smaller model as the regularizer without employing dropout in that network, opting instead to reduce its complexity by modifying the number of layers or nodes? Btw, does the main model use any regularization (weight decay, etc.)?

2. Given that the regularizer network does not contribute to gradient flow and essentially remains untrained, were different weight and bias initialization methods explored for the regularizer network, and if so, what were the outcomes?

3. It would be valuable to assess the performance and generalization capabilities of this approach on a wider range of datasets. Specifically, results on datasets like CIFAR-C, ImageNet-C, and domain generalization datasets such as VLCS would provide a broader perspective on the model's effectiveness.

4. This work reminds me another paper (https://arxiv.org/abs/2207.01548) that focuses on regularization by using an additional model without batch normalization (BN). It might be interesting to delve into the connections and distinctions between these two approaches. This makes me more think of point #1 above.

**Questions:**

Please see Weaknesses section

---

> ### Author Response · Authors · 2023-11-15
> **Official Response to Reviewer 2qPD 1/2**
>
> > W1: 1. Were there any experiments conducted using a smaller model as the regularizer without employing dropout in that network, opting instead to reduce its complexity by modifying the number of layers or nodes?
>
> Answer: We appreciate the reviewer's suggestion regarding model regularization for the dense model. However, we clarify that adopting a smaller model may not be reasonable for two reasons:
> 1. **Benefits we aim to leverage**: The primary motivation of this study is to harness benefits such as the robust generalization ability inherent in dropout models. We posit that only larger models have the potential to replicate these benefits akin to those observed in dropout models.
> 2. **Cost consideration:** Implementing a larger model for regularization, with the objective of achieving robust generalization ability, will invariably incur significant memory and computational costs.
>
> In summary, while larger models may emulate the advantages of dropout models, they also bring substantial memory and computational costs. MoReDrop addresses these issues through regularization, implemented via a shared-parameter sub-model within the training dense model.
>
> > W1: Btw, does the main model use any regularization (weight decay, etc.)?
>
> Answer: Yes, our implementation closely follows the official open-source version, and we have maintained the same configuration settings as the original implementation. Specifically, we have incorporated weight decay in the implementation of the ViT-B setting, mirroring the original approach. Additionally, we use batch normalization in ResNet, as proposed in the original code. Further details about the data augmentation methods adopted in our work can be found in Appendix B.
>
> > W2: Given that the regularizer network does not contribute to gradient flow and essentially remains untrained, were different weight and bias initialization methods explored for the regularizer network, and if so, what were the outcomes?
>
> Answer: We appreciate the reviewer's query, as it contributes to a deeper understanding of MoReDrop. In response, conducting this ablation study with MoReDrop is not feasible due to the shared nature of the regularizer network with the dense training model. Initializing two separate networks—one for training and another for regularization—contradicts our established, such as the cost consideration, as detailed in response to W1.
>
> > W3: It would be valuable to assess the performance and generalization capabilities of this approach on a wider range of datasets. Specifically, results on datasets like CIFAR-C, ImageNet-C, and domain generalization datasets such as VLCS would provide a broader perspective on the model's effectiveness.
>
> Answer: Thank the reviewer for your valuable suggestion regarding the evaluation of our model's robustness and generalization capabilities across a wider range of datasets. In line with your recommendations, we have extended our experimental framework to include additional datasets such as CIFAR-10-C, ImageNet-C (blur and digital) on WRN-28-10, and conducted an analysis under the PGD attack.
>
> Due to time constraints, we standardized the experimental setup for all datasets by fixing $(p,\alpha)=(0.3,1)$ for CIFAR-10-C and PGD attack scenarios, and $(p,\alpha)=(0.1,1)$ for ImageNet-C. This decision allowed us to focus on comparing the effectiveness of our method, MoReDrop, across different datasets and conditions while maintaining a consistent experimental environment.
>
> | method |CIFAR-10 | PGD|CIFAR-10-C |
> |-----|-----|-----|-----|
> | WRN-28-10 |94.65 | 23.83 |76.27   |
> | +MoReDrop |**94.81** | **28.53** |**77.01**   |
>
> | method |ImageNet |ImageNet-blur|ImageNet-digital|
> |-----|-----|-----|-----|
> | ViT-B |84.01 | 51.86 |61.58 |
> | +MoReDrop |**84.47**|**52.18**  | **62.30**|
>
> The tables presented above demonstrate that MoReDrop significantly improves model generalization and robustness. Notably, we enhanced the PGD accuracy without implementing any specific defense mechanisms. These results were attained without fine-tuning, relying solely on empirically derived hyperparameter settings. We believe that a proper parameter setting can further improve the robustness and generalization.

---

> > ### Comment · Reviewer_2qPD · 2023-11-18
> > **thank you for the response**
> >
> > Thank you for the response.
> >
> > - Regarding #1, leveraging a smaller network (without dropout) would actually reduce computational costs. Instead of using a network with dropout, simply make the network smaller and don't use dropout. This is, in fact, less computationally expensive than the dropout model.
> >
> > - Thank you for including the ImageNet-C results. Is there a specific reason for excluding other ImageNet-C sets?
> >
> > - Regarding PGD, since you use dropout, there might be gradient masking due to the randomness, which could compromise the reliability of the defense. Please refer to this paper for more details: https://arxiv.org/abs/1802.00420.

---

> > > ### Author Response · Authors · 2023-11-19
> > > **Official Response to the Further Questions to Reviewer 2qPD**
> > >
> > > > smaller model is less computationally expensive than the dropout model in MoReDrop.
> > >
> > > In response to W1, our initial assertion is that the use of smaller models for regularization is not entirely logistic for our motivation with the potential benefits, such as improved generalization, which are typically provided by larger models. However, larger models Inevitably introduce memory cost and computation.
> > >
> > > Furthermore, even if smaller models were to provide certain benefits such as generalization ability (a proposition we question with an extreme degree of confidence), the computational cost may still be greater than the dropout model from MoReDrop. Essentially, the dropout model requires only forward passes, contrasting with smaller models that necessitate both forward and backward passes, where the backward pass typically imposes an extra computational burden. Nevertheless, we would like to agree with the assertion from reviewer 2qPD that the reduced computational cost in the forward pass may sufficiently offset the cost of the backward pass. However, it is important to underscore that the implementation of a smaller model for the regularizer is not entirely logical regarding our motivation.
> > >
> > > > Is there a specific reason for excluding other ImageNet-C sets?
> > >
> > > Thank you for your inquiry regarding the selection of ImageNet-C subsets. Initially, we chose ImageNet-blur (7.1GB) and ImageNet-digital (7.8GB) due to their manageable sizes for an efficient research process, given the time constraints of the rebuttal period.
> > >
> > > Acknowledging your concerns, we extended our evaluations to include ImageNet-extra (15.8GB), ImageNet-noise (22.6GB), and ImageNet-weather (12.8GB). The results are presented below:
> > >
> > > |method|ImageNet|ImageNet-blur|ImageNet-digital|ImageNet-extra|ImageNet-noise|ImageNet-weather|ImageNet-C|
> > > |---|---|---|---|---|---|---|---|
> > > |ViT-B|84.01|51.86|61.58|61.20|48.69|59.42|56.55|
> > > |+MoReDrop|**84.47**|**52.18**|**62.30**|**62.61**|**51.03**|**59.84**|**57.59**|
> > >
> > > In our forthcoming manuscript, we intend to incorporate additional experimental results and analyses to emphasize the superiority of MoReDrop in those tasks that necessitate robustness and the ability to generalize. This will serve to enhance the overall strength and validity of MoReDrop.
> > >
> > > > Regarding PGD, since you use dropout, there might be gradient masking due to the randomness, which could compromise the reliability of the defense.
> > >
> > > Thanks for sharing your thoughts regarding the potential in MoReDrop's defense against PGD attacks.MoReDrop is primarily focused on optimizing the dense model, without active updates to sub-models (dropout models). This design choice significantly reduces the likelihood of intentional gradient masking during the backpropagation process, so we believe that MoReDrop's effectiveness against PGD is not attributable to this phenomenon as discussed in [Athalye et al. (2018)](https://arxiv.org/pdf/1802.00420.pdf).
> > >
> > > Instead, as indicated in [Wang et al., 2018](https://arxiv.org/pdf/1809.05165.pdf), MoReDrop's enhanced defense is likely due to its improved generalization, resulting in a distribution of gradients that is shorter and fatter during attacks. To provide a clearer understanding, we have plotted the gradient distributions for both the baseline and MoReDrop [here](https://imgur.com/a/XVTKxL8) (link anonymized for review).
> > >
> > > While MoReDrop shows potential in defending against PGD attacks, we acknowledge that combining it with other defense mechanisms could further strengthen its efficacy. As our current focus is not primarily on defense and attack strategies, we look forward to exploring these aspects in future work. We appreciate the broader perspective on the strength of MoReDrop from reviewer 2qPD.

---

> > > ### Author Response · Authors · 2023-11-22
> > > **Official Comments from Authors**
> > >
> > > Dear reviewer, The deadline is approaching. We would appreciate it if you could reply to our rebuttal. We are sincerely looking forward to further discussions to address the reviewer's concerns to our best. Thanks!

---

> ### Author Response · Authors · 2023-11-15
> **Official Response to Reviewer 2qPD 2/2**
>
> > W4: This work reminds me of another paper (https://arxiv.org/abs/2207.01548) that focuses on regularization by using an additional model without batch normalization (BN). It might be interesting to delve into the connections and distinctions between these two approaches. This makes me think of point #1 above.
>
> Answer: we would like to share our grateful appreciate for this related work.  We delineate the Similarities and distinctions between our work and the referenced paper ([Taghanaki et al., 2022](https://arxiv.org/pdf/2207.01548.pdf)).
>
> **Similarities**: Both our research and [Taghanaki et al., 2022](https://arxiv.org/pdf/2207.01548.pdf) aim to augment the generalization and robustness of models. Notably, the training process in both instances involves the interaction of two models.
>
> **Distinctions**
> 1. **Motivation**: [Taghanaki et al., 2022](https://arxiv.org/pdf/2207.01548.pdf) focus on addressing a problem specific to the Batch Normalization's (BN). In contrast, the motivation of our work focuses on how to leverage the benefits of dropout models and avoid the model distributional shift.
> 2. **Methods**:  [Taghanaki et al., 2022](https://arxiv.org/pdf/2207.01548.pdf) adheres to the standard distillation training paradigm to solve the related work in BN, but MoReDrop has no relationship with the distillation paradigm but a sub-model with dropout.
> 3. **Cost**: The distillation training paradigm requires the teacher and student models with different parameters, it requires extra memory cost but MoReDrop does not because of the shared parameter between the dropout model and the main dense model.
> 4. **Scalability**: The problem that [Taghanaki et al., 2022](https://arxiv.org/pdf/2207.01548.pdf) aim to mitigate specific to the BN, yet not the same scenarios in other normalization methods, such as group normalization. However, our method is scalable and compatible with various dropout variants to further improve the performance.

---

### Official Review · Reviewer_7aV5 · 2023-10-31

**Soundness:** 3 good
**Presentation:** 3 good
**Contribution:** 2 fair
**Rating:** 5
**Confidence:** 4

**Summary:**

The paper introduces MoReDrop, a novel solution addressing the distributional shift problem between training and inference when using dropout regularization for neural networks. To tackle the issue, MoReDrop introduces a novel loss function and adds a regularization term derived from predictions with dropout sub-models. Additionally, MoReDropL, a more computationally efficient variant, employs dropout only at the last layer, balancing generalization and computational cost. The experiments conducted on various benchmarks demonstrate the scalability and efficiency of MoReDrop and MoReDropL.

**Strengths:**

- Well-Structured Presentation: The paper is well-structured, with a clear introduction, detailed methodology, and comprehensive experimental results. It presents its findings in a logical and accessible manner.

- Comprehensive Experimental Validation: The paper backs its claims with thorough experimental evaluations conducted across multiple benchmarks and domains.

- Clear Problem Statement: The paper clearly defines the problem of distributional shift and the limitations of existing methods.

- Practical Implications: The solutions presented in the paper have practical implications, potentially leading to more stable and robust deep learning models in various applications.

**Weaknesses:**

- Is there any theoretical and/or empirical analysis on the effect of adding the proposed regularization term to the model distribution shift $\mathcal{G}$? I feel like there is insufficient justification/explanation on how the proposed algorithm can reduce the model distribution shift. Is it really helping by reducing such a shift? In general, it would be great if the authors can make connections better between equation 1 and the subsequent equations/derivations.

-  I am not fully convinced by the use of equation 4 for the loss function. It would be good if additional empirical analysis can be done on the effect of different choices of $g(\cdot)$ have on the results.

- An additional ablation study on the quality of sub-model predictions have during the course of entire training would be very helpful in facilitating the understanding on exactly how the proposed method is helping regularizing the model.

- The proposed method feels very similar to the use of "exponential moving average (EMA) predictions" in semi-supervised learning [1]. I wonder if the proposed algorithm is just similar things to that. It also sounds similar to another line of research on self-distillation [2]. A baseline comparison and a literature review along these two lines of work would be helpful.

[1] Antti Tarvainen and Harri Valpola. Mean teachers are better role models: Weight-averaged consistency targets improve semi-supervised deep learning results. In Advances in neural information processing systems, pages 1195–1204, 2017.

[2] Zhang, Linfeng, et al. "Be your own teacher: Improve the performance of convolutional neural networks via self distillation." Proceedings of the IEEE/CVF international conference on computer vision. 2019.

**Questions:**

- Why does MoReDropL perform better than MoReDrop from the model distribution shift perspective?

- Any additional explanation on why the proposed method can perform as well/better on high dropout-rate? It seems like higher dropout rate probably lead to less accurate predictions.

---

> ### Author Response · Authors · 2023-11-15
> **Official Response to Reviewer 7aV5 Regarding Weaknesses 1/2**
>
> > W1: Is there any theoretical and/or empirical analysis on the effect of adding the proposed regularization term to the model distribution shift $\mathcal{G}$? I feel like there is insufficient justification/explanation on how the proposed algorithm can reduce the model distribution shift. Is it really helping by reducing such a shift? In general, it would be great if the authors could make connections better between Equation 1 and the subsequent equations/derivations.
>
> Answer: We thank the reviewer for raising this important question and we have revised this paper for further clarification. In response, we clarify that the key contributor to the model distributional shift is the active updating of dropout (sub-) models. In other words, the gap exists and can be represented by $\mathcal{G}$ as long as the dropout model performs the gradient backpropagation.
>
> However, MoReDrop actively updates only the dense model; the sub-model is passively updated due to shared attributes. This maintains the model consistency, i.e., the dense model, during training and evaluation, as so **the distributional shift is zero!** It has no relationship between $\mathcal{G}$ because we never perform gradient backpropagation on the shared-parameter sub-model. Besides, the regularizer retains dropout benefits in the dense model. We also present a clear comparison between MoReDrop and previous methods in Appendix Table 4:
>
> | Algorithms | Gradient back-propagation | Dense model update | Sub-model update | Regularizer | Distribution shift |
> |------------|---------------------------|--------------------|------------------|-------------|-------------------|
> | FD [[Zolna et al., 2018](arxiv.org/pdf/1711.00066.pdf)] | Sub-model | Passive | Active | Sub-to-sub | $\mathcal{G}$ |
> | R-Drop [[Liang et al., 2021](arxiv.org/pdf/2106.14448.pdf)] | Sub-model | Passive | Active | Sub-to-sub | $\mathcal{G}$ |
> | WordReg [[Xia et al., 2023]()] | Sub-model | Passive | Active | Sub-to-sub | $\mathcal{G}$ |
> | ELD [[Ma et al., 2017](arxiv.org/pdf/1609.08017.pdf)] | Sub-model | Passive | Active | Dense-to-sub | $\mathcal{G}$ |
> | MoReDrop (ours) | Dense model | Active | Passive | Dense-to-sub | 0 |
>
> > W2: I am not fully convinced by the use of equation 4 for the loss function. It would be good if additional empirical analysis could be done on the effect of different choices of $g(\cdot)$  have on the results.
>
> Answer: We greatly appreciate your assistance in strengthening our paper with regard to the loss function $g$. In response, our loss function ( $g(x)=\frac{e^x-1}{e^x+1}=2\cdot(sigmoid(x)-\frac{1}{2})$) and we clarify two main strengths:
> 1. The bound nature of this loss function between (-1, 1).
> 2. The superior in contrast to loss functions of other dropout regularization methods.
> However, it is important to acknowledge the possibility of other, more effective loss functions, which we aim to explore in future research.
>
> **Whether the bound nature between (-1, 1) efficient in the experimental view?**  For practicality and simplicity in implementation, we adopt ResNet-18 as our model backbone. We configure four distinct experimental scenarios: ResNet-18-2D, which incorporate 2 dropout layers.
>
> | $g(\cdot)$  | p=0.1 | p=0.3 |p=0.5|p=0.7 |p=0.9|
> |-----|-----|-----|-----|-----|-----|
> | $g(x)=x$    | 95.22    |  95.31  | 95.13  | 95.04 | 90.07  |
> | $g(x)=x^2$     | 95.42    | 95.51   | 95.33   | 95.23 | 94.23  |
> | $g(x)=\frac{e^x-1}{e^x+1}$    |  **95.56**   | **95.80**   | **95.72**  |  **95.47** | **95.23**  |
>
> The tables above reveal distinct performance trends across different $g(\cdot)$ functions. Specifically,$g(x)=\frac{e^x-1}{e^x+1}$ outperforms other functions in all settings. With the increasing dropout rate, we can find the performance gap between $g(x) = x^2$ and $g(x) = \frac{e^x - 1}{e^x + 1}$ is also increasing, where $g(x) = \frac{e^x - 1}{e^x + 1}$ exhibits superior robustness in these extreme conditions.
>
> **$\alpha \cdot g(x)$ or $g(\alpha \cdot x)$?** We then perform an examination on these two forms of the loss function with respect to the location of $\alpha$, where the second one is also bounded between (-1, 1). To this end, an experiment was conducted using ResNet-18-4D with $p = 0.5$ shown in the following table:
>
> | $g(\cdot)$  | $\alpha=0.1$ | $\alpha=0.5$ |$\alpha=1$|$\alpha=2$ |$\alpha=5$ |$\alpha=10$ |
> |-----|-----|-----|-----|-----|-----|-----|
> | $\alpha \cdot g(x)$   | **95.55**    | 95.81   |  95.61 | **95.48** | 94.28|73.80|
> | $g(\alpha \cdot x)$  | 95.52 | **95.83**   |  **95.66** | **95.48** |**95.36** |**95.22**|
>
> We observe that the bounded loss function  $g(\alpha \cdot x)$ shows significant robustness in the extreme choices of $\alpha$, offering more versatility in addressing a range of issues. We acknowledge the limit discrepancy for the optimal solution between those two functions, i.e., $\alpha=0.5$. However, it is important to note that the optimal $\alpha$ may exhibit dramatic changes in some extreme tasks.

---

> ### Author Response · Authors · 2023-11-15
> **Official Response to Reviewer 7aV5 Regarding Weaknesses 2/2**
>
> > W3: An additional ablation study on the quality of sub-model predictions during the course of the entire training would be very helpful in facilitating the understanding of exactly how the proposed method is helping regularize the model.
>
> Answer: We appreciate the reviewer's input which has enhanced our paper's clarity. In Section 4.3, we present a thorough analysis of the loss function (the proxy to the prediction), comparing MoReDrop and R-Drop, two standard algorithms in this field. Our loss analysis also reveals that MoReDrop maintains the model generalization ability, despite imposed constraints. In contrast, R-Drop suffers from model distributional shift and compromised generalization, leading to its lower performance compared to MoReDrop.
>
> Furthermore, we here want to highlight that the primary loss function from the dense model and gradient backpropagation solely to the dense model ensures model configuration consistency for training and evaluation. In other words, this design allows MoReDrop without model distributional shift, a clear comparison is shown in Appendix Table 4 and the first answer. The regularization in MoReDrop provides the dropout model benefits.
>
> > The proposed method feels very similar to the use of "exponential moving average (EMA) predictions" in semi-supervised learning [1]. I wonder if the proposed algorithm is just similar things to that. It also sounds similar to another line of research on self-distillation [2]. A baseline comparison and a literature review along these two lines of work would be helpful.
>
> Answer: We thank the reviewer for referring to more related work. In response, we think those two related work shows a significant discrepancy to our work,  especially the motivation, and the comparison is redundant. We present the reasons one by one:
>
> With [Tarvainen et al.,2017](https://proceedings.neurips.cc/paper/2017/file/68053af2923e00204c3ca7c6a3150cf7-Paper.pdf)
> 1. **Motivation**: [Tarvainen et al.,2017](https://proceedings.neurips.cc/paper/2017/file/68053af2923e00204c3ca7c6a3150cf7-Paper.pdf) aim to mitigate the limitation of Temporal Ensembling in semi-supervised learning. In contrast, the motivation of our work is to mitigate the model distributional shift problem with dropout.
> 2. **Method**: [Tarvainen et al.,2017](https://proceedings.neurips.cc/paper/2017/file/68053af2923e00204c3ca7c6a3150cf7-Paper.pdf) improve the generalization ability by different data sources, i.e., the noisy input information. In contrast, our method preserves the generalization ability from the model perspective by the regularization of the dropout sub-model.
> 3. **Memory cost:** [Tarvainen et al.,2017](https://proceedings.neurips.cc/paper/2017/file/68053af2923e00204c3ca7c6a3150cf7-Paper.pdf) adopts the idea of distillation with teacher and student models, which requires extra memory to store these two models because they are not shared parameter.
>
> With [Zhang et al.,2019](https://openaccess.thecvf.com/content_ICCV_2019/papers/Zhang_Be_Your_Own_Teacher_Improve_the_Performance_of_Convolutional_Neural_ICCV_2019_paper.pdf)
> 1. **Motivation**: [Zhang et al.,2019](https://openaccess.thecvf.com/content_ICCV_2019/papers/Zhang_Be_Your_Own_Teacher_Improve_the_Performance_of_Convolutional_Neural_ICCV_2019_paper.pdf)  advocate a self-distillation approach to enhance the performance on a smaller network, i.e., some modules from the entire model. However, the motivation of MoReDrop is to address the model distribution shift with the dense model and its shared-parameter sub-model, where these two models are the entire model with the same model size.
> 2. **Method**: [Zhang et al.,2019](https://openaccess.thecvf.com/content_ICCV_2019/papers/Zhang_Be_Your_Own_Teacher_Improve_the_Performance_of_Convolutional_Neural_ICCV_2019_paper.pdf)  enhance shallow module predictions (a smaller model) using deep modules, whereas MoReDrop uses two equally sized models. Note that the sub-model definition in those two works is significantly different.
> 8. **Memory cost**: [Zhang et al.,2019](https://openaccess.thecvf.com/content_ICCV_2019/papers/Zhang_Be_Your_Own_Teacher_Improve_the_Performance_of_Convolutional_Neural_ICCV_2019_paper.pdf)  necessitate additional training modifications such as supplementary softmax or classifier layers, which incur an additional memory cost. In contrast, our MoReDrop method does not require such adjustments, as we did not introduce any extra parameters during the training process.
>
> To summarize, there is a notable divergence between our work and the two referenced studies, particularly in terms of motivation. Consequently, we consider comparisons with these studies to be less feasible and superfluous.

---

> > ### Comment · Reviewer_7aV5 · 2023-11-17
> >
> > I would like to thank the authors for addressing my concerns regarding the paper. The reason why I brought in Tarvainen et al.,2017 and Zhang et al.,2019 into the discussion is as follows. Based on my understanding, it sounds like the explanations/argument for the observed improvement is from the "distribution shift between training and testing" perspective. However, another way to look at what the proposed method is doing is that, it is basically using an easy-to-obtain and less expressive model as a way to regularize the full dense model. In the case of MoReDrop, such a "teacher model" is the dropout model. However, other alternatives exist. For example, we can also use a temporal ensemble as proposed by Tarvainen et al.,2017. There have been numerous approaches proposed to guide/regularize the training of NNs with some form of "teacher model", and I am not trying to ask for an exhaustive comparison against all of these prior works. However, I am not fully convinced by the motivation of the proposed algorithm from the "distribution shift between training and testing" perspective, and feel that the connection between the distribution shift problem and the proposed method needs to be strengthened further. This is the case especially since, to the best of my understanding, according to Algorithm 1, only one sub-model is sampled during each iteration of training, instead of approximating the expectation over dropout masks according to Eqn 4. I would really appreciate it if the authors of the paper can address the concerns and the discrepancy between Algorithm 1 and Eqn 4. Thanks!

---

> ### Author Response · Authors · 2023-11-15
> **Official Response to Reviewer 7aV5 Regarding Questions**
>
> > Q1: Why does MoReDropL perform better than MoReDrop from the model distribution shift perspective?
>
> Answer: We would like to share our gratitude to the reviewer for this question. We find the superior performance on MoReDropL on the GLUE tasks. We want clarify that both MoReDrop and MoReDropL have no model distributional shift, as we highlighted before. However, we would like to provides two potential directions for this phenomenon:
> 1. **The loss of plasticity:** The further training has limited impact on most of neurons and it may even cause the plasticity loss of neural networks ([Achille et al., 2017](https://arxiv.org/abs/1711.08856), [G Zilly et al, 2023](https://www.research-collection.ethz.ch/handle/20.500.11850/578035))
> 2. **Preservation of Pre-Trained Features in the initial layers**: Pre-trained models have learned complex features from large datasets. MoReDrop, by applying dropout in all layers, risks disrupting these well-established features, especially in the initial layers where fundamental, general-purpose patterns are encoded. In contrast, MoReDropL preserves these features by limiting dropout to the last layer. A recent work demonstrate that the efficient of last-layer pre-training ([Kirichenko et al., 2023](http://arxiv.org/abs/2204.02937))
>
> > Q2: Any additional explanation on why the proposed method can perform as well/better on high dropout-rate? It seems like higher dropout rate probably lead to less accurate predictions.
>
> Answer: We thank the reviewer for this question, which further enhances the understanding of MoReDrop. We here clarify for two main reasons:
>
> **The training paradigm of MoReDrop**. In MoReDrop, the primary gradient update mechanism is derived from traditional dense training, distinct from the influence of the regularizer. The regularizer integrates benefits of dropout models into the dense training framework. The parameter $\alpha$ is pivotal in modulating the extent to which the regularizer, denoted as $\mathcal{R}$, impacts the gradient update in the dense model. In scenarios with high dropout rates, a reduced $\alpha$ value minimizes the impact from the regularizer, preserving the performance of the dense model that primarily relies on conventional training methods sans dropout.
>
> This approach aligns with findings from our ablation study in Appendix C.4.1, indicating the importance of lower $\alpha$ values in sustaining model performance under increased dropout rates. In contrast, a higher $\alpha$ is advantageous in lower dropout scenarios to borrow advantages from dropout models, leveraging the reliable predictions of the dropout model.
>
> **The bound nature of $\mathcal{R}$.** The inherent boundedness of the regularizer $\mathcal{R}$ facilitates constraining the gap to less than 1, coupled with the weight $\alpha$, exerts limited influence on the optimization of the dense model.
>
> In summary, in high dropout rates, the performance is retained mostly from the dense model by setting a smaller $\alpha$, this inherits from the unique training paradigm of MoReDrop and the bound nature of the regularizer.

---

> ### Author Response · Authors · 2023-11-18
>
> Thanks for further questions! We want to break out the last comments into 2 points for better clarification.
>
> > Another explanation with a less expressive model as a way to regularize the full dense model: teacher-student perspective.
>
> We would like to clarify that there may be a facial connection between teacher-student perspective with MoReDrop, but they are completely different with deeper understanding. From our best understanding, there are several key properties of the teacher-student model: (1). teacher model is expected to outperform or be more expressive compared to the student model, at least in the beginning. (2). with the promise of the first property, the teacher model is typically bigger than the student model without sharing the whole network (3). with the promise of the first property, the learning process of teacher and student models is asynchronous, i.e., learn a good teacher model first.
>
> However, MoReDrop cannot be simply viewed as the teacher-student model. The motivation of MoReDrop can be also broken into two points: (a). the primary loss optimization only on the dense model, ensuring the consistency of the model for training and testing. (b). The regularizer retains dropout benefits, e.g., generalization. It means, the performance of the shared-parameter dropout model is anchored to the dense model: the dropout model is not always better than the dense model regarding the performance, especially at the beginning and high dropout rates; it is synchronously trained with the dense model; and it share the same whole network.
>
> The dropout model benefits, especially for the generalization ability, can not be to achieved by the regularization on a less expressive model (teacher model as the reviewer said) but a more expressive model. However, it is not elegant and feasible in terms of computation and memory consideration by using a larger model.
>
> > The discrepancy between Algorithm 1 and Eqn 4.
>
> Thank the reviewer for pointing out this consideration. Reviewer 7aV5 shares the same consideration with reviewer 8mBK for the same problem but in a different perspective. We would like to highly recommend reviewer 7aV5 to synchronize our discussion with reviewer 8mBK.
>
> In response, dropout aims at, **simultaneously and jointly**, training an ensemble of exponentially many neural networks (one for each conﬁguration of dropped units) while sharing the same weights/parameters. (see almost the same context in [Liang et al. (2021)](http://arxiv.org/abs/2106.14448) (page 1, Section 1)  [Ma et al., 2017](https://openreview.net/forum?id=rkGabzZgl) (page 2, Section 2.2)). In other words, despite the presence of only a single explicit dropout model during training at every training step, this effectively represents implicitly an ensemble of numerous dropout models due to the variability introduced by the set of dropout random variables $S$, so you can safely ignore the expectation term. More evidence can also be found in Eq (2) in [Liang et al. (2021)](http://arxiv.org/abs/2106.14448) with the expectation term.

---

> > ### Comment · Reviewer_7aV5 · 2023-11-20
> >
> > I would like to thank the authors for addressing my concerns about the discrepancy between algorithm 1 and Eqn 4. In my opinion, the authors should also update the paper accordingly to address the discrepancy, and mention the relevant prior works for their contributions (i.e. Liang et. al 2012 and Ma et al. 2017) in the section to address this discrepancy.
> >
> > Regarding connection to teacher-student training: prior work [1] has demonstrated that the teacher network can be much shallower and less expressive than the student networks, and that the benefits of knowledge distillation come actually from "regularization" effect. As such, from this perspective, I still believe that the connection between the proposed method and the teacher-student perspective is deeper than it sounds like.
> >
> > [1] Yuan, Li, et al. "Revisiting knowledge distillation via label smoothing regularization." Proceedings of the IEEE/CVF Conference on Computer Vision and Pattern Recognition. 2020.

---

> ### Author Response · Authors · 2023-11-20
> **Official Response to Reviewer 7aV5**
>
> > Suggestions on the discrepancy between Eq 4 and Algorithm 1.
>
> Thanks for your suggestion to avoid further misinterpretation. We have provided this in our last manuscript with "This procedure implicitly establishes a parameter-sharing ensemble mode (page 2, Section 3.1)" and we have highlighted it again in Section 3.2 with the algorithm summary (newly uploaded manuscript with red color).
>
> > The connection between the student-teacher perspective
>
> We express our deepest gratitude to reviewer 7aV5 for highlighting the connection. The incorrect properties of the teacher-student paradigm in the last response is a traditional perspective and we would apologize for it due to our limited expertise in this area.  Given the time constraints of the rebuttal period, we assure that a thorough examination of related work will be included in our final manuscript.
>
> Putting aside their connection and backing to the initial question: Can the dropout model be replaced with a smaller model? Our response is negative, rooted in our pursuit of generalization benefits derived from dropout (or larger models), as well as the tractable computation burden and memory cost consideration. However, there may exist some contradictory results, similar to Yuan et al., 2020. We are open to conducting additional experiments, i.e., regularization on a smaller dense model, if reviewer 7aV5 deems them necessary and crucial for this paper.

---

> > ### Author Response · Authors · 2023-11-22
> > **Official Comments from Authors**
> >
> > Dear reviewer, The deadline is approaching. We would appreciate it if you could reply to our rebuttal. We are sincerely looking forward to further discussions to address the reviewer's concerns to our best. Thanks!

---

### Meta-Review · Area_Chair_9fxg · 2023-12-05

**Metareview:**

The authors attempt to address the distribution shift experienced by dropout-regularized models between training and evaluation. The proposed approach, MoReDrop, combines the dense models loss function with a regularization term. Results are presented on image and text classification/regression settings. The reviewers appreciated the clear presentation of the paper and the novelty of the approach, however, concerns were raised regarding the magnitude of improvement, the underlying reasons for improved performance, and various other aspects of the paper.

**Justification For Why Not Higher Score:**

The paper presents an interesting approach, however, given the confusion over the theoretical justifications of the approach and the marginal empirical improvements, this paper is not ready for publication.

**Justification For Why Not Lower Score:**

N/A

---

### Decision · Program_Chairs · 2024-01-16

Reject